# Simulating multiple faceted variability in single cell RNA sequencing

Xiuwei Zhang [1,2,4], Chenling Xu [1,4] & Nir Yosef [1,2,3]

The abundance of new computational methods for processing and interpreting transcriptomes at a single cell level raises the need for in silico platforms for evaluation and validation. Here, we present SymSim, a simulator that explicitly models the processes that give rise to data observed in single cell RNA-Seq experiments. The components of the SymSim pipeline pertain to the three primary sources of variation in single cell RNA-Seq data: noise intrinsic to the process of transcription, extrinsic variation indicative of different cell states (both discrete and continuous), and technical variation due to low sensitivity and measurement noise and bias. We demonstrate how SymSim can be used for benchmarking methods for clustering, differential expression and trajectory inference, and for examining the effects of various parameters on their performance. We also show how SymSim can be used to evaluate the number of cells required to detect a rare population under various scenarios.

[1] Department of Electrical Engineering and Computer Sciences, Center for Computational Biology, UC Berkeley, Berkeley, CA 94720, USA. [2] Ragon Institute of Massachusetts General Hospital, MIT and Harvard, Cambridge, MA 02139, USA. [3] Chan-Zuckerberg Biohub, San Francisco, CA 94158, USA. [4] These authors contributed equally: Xiuwei Zhang, Chenling Xu. Correspondence and requests for materials should be addressed to N.Y. (email: niryosef@berkeley.edu)

The advent of single cell RNA sequencing has led to a surge of computational and statistical methods for a range of analysis tasks. Some of the methods or the tasks that they perform have originated from bulk sequencing analysis, while others address opportunities (e.g., identification of new cell states[1,2]) or technical limitations (e.g., limited sensitivity[3,4]) that are idiosyncratic to single cell genomics[5,6]. While these computational methods are often based on reasonable assumptions it is difficult to compare them to each other and assess their performance without gold standards. One approach to address this is through simulations[7–12].

Existing simulation strategies (summarized by Zappia et al.[13]) rely primarily on fitting distributional models to observed data and then drawing from these distributions. While the resulting models provide a good fit to observed data, their parameters are often abstract and do not directly correspond to the actual processes that gave rise to the observations. This leaves an important unaddressed problem in designing and using a simulator: the need to modulate and then study the effects of specific aspects of the underlying physical processes, such as the efficiency of mRNA capture, the extent of amplification bias (e.g., by changing the number of PCR cycles, or by using unique molecular identifiers [UMI]), and the extent of transcriptional bursting. To address this, we present SymSim (Synthetic model of multiple variability factors for Simulation), a software for simulation of single cell RNA-Seq data. SymSim explicitly models three of the main sources of variation that govern single cell expression patterns[2]: allele intrinsic variation, extrinsic variation, and technical factors (Fig. 1 and Supplementary Fig. 1). SymSim provides the users with knobs to control various parameters at these three levels. First, we generate true numbers of molecules using a kinetic model, which allows us to adjust allele intrinsic variation and the extent of burst effect; second, we provide an intuitive interface to simulate a subpopulation structure, either discrete or along a continuum, through specification of cluster-trees, which define a low-dimensional manifold from which the transcriptional kinetics is determined for every gene and every cell; third, we simulate the main stages of the library preparation process and let users control the amount of variation stemming from these steps, such as capture efficiency, amplification bias, varying sequencing depth, and batch effect. Importantly, through this modeling scheme, SymSim recapitulates properties of the data (e.g., high abundance of zeros or increased noise in non-UMI protocols) without the need to explicitly force them as factors in a distributional model.

We demonstrate the utility of SymSim in two types of applications. In the first example, we use it to evaluate the performance of algorithms. We focus on the tasks of clustering, differential expression and trajectory inference, and test a number of methods under different simulation settings of biological separability and technical noise. In the second example, we use SymSim for the purpose of experimental design, focusing on the question of how many cells should one sequence to identify a certain subpopulation.

## Results

**Allele intrinsic variation**. The first knob for controlling the simulation allows us to adjust the extent to which the infrequency of bursts of transcription adds variability to an otherwise homogenous population of cells. We use the widely accepted two-state kinetic model, in which the promoter switches between an on and an off states with certain probabilities[14,15]. We use the notation $k_{on}$ to represent the rate at which a gene becomes active, $k_{off}$ the rate of the gene becoming inactive, $s$ the transcription rate, and $d$ the mRNA degradation rate. For simplicity, and following previous work, we fix $d$ to constant value of $1^{14,16}$ and consider the other three parameters relative to $d$. Since RNA sequencing

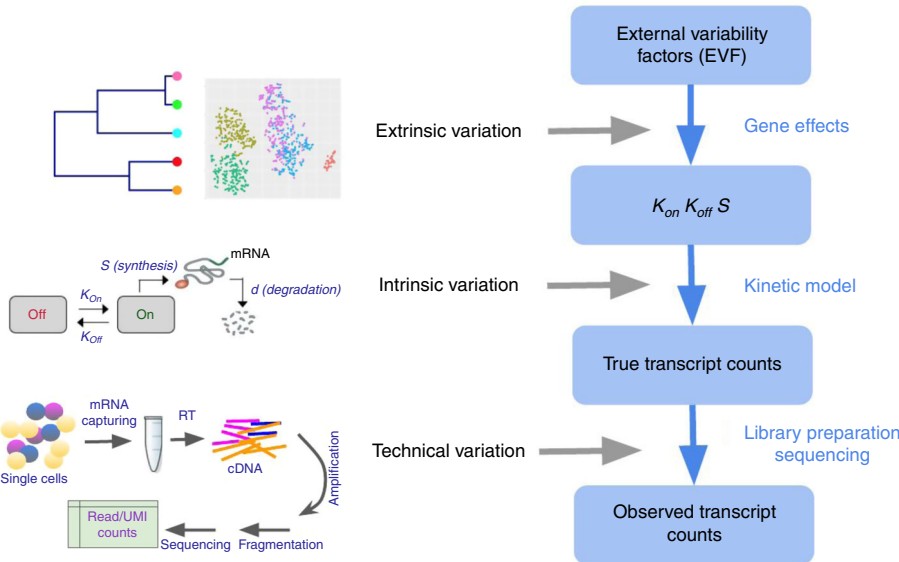

**Fig. 1** Overview of SymSim. The true transcript counts, which are the number of molecules for each transcript in each cell at the time of analysis, are generated through the classical promoter kinetic model with parameters: promoter *on* rate ($k_{on}$), *off* rate ($k_{off}$) and RNA synthesis rate ($s$). The values of the kinetic parameters are determined by the product of gene-specific coefficients (termed gene effects) and cell-specific coefficients. The latter set of coefficients is termed extrinsic variability factors (EVF), and it is indicative of the cell state. The expected value of each EVF is determined in accordance to the position of the cell in a user-defined tree structure. The tree dictates the structure of the resulting cell–cell similarity map (which can be either discrete or continuous) since the distance between any two cells in the tree is proportional to the expected distance between their EVF values. For homogenous populations (represented by a single location in the tree), the EVFs are drawn *iid* from a distribution whose mean is the expected EVF value and variance is provided by the user. From the true transcript counts we explicitly simulate the key experimental steps of library preparation and sequencing, and obtain observed counts, which are read counts for full-length mRNA sequencing protocols, and UMI counts, otherwise

provides a single snapshot of the transcriptional process, we resort to assuming that the cells are at a steady state, and thus that the resulting single-cell measurements are drawn from the stationary distribution of the two-state kinetic model. Since $d$ is fixed, we are able to express the stationary distribution for each gene analytically using a Beta-Poisson mixture[17] (Methods).

The values of the kinetic parameters ($k_{on}$, $k_{off}$, and $s$) for each gene in each cell are first calculated using a product of cell-specific and gene-specific factors, then adjusted by the parameter distributions estimated from experimental data (Fig. 2a, Methods). Specifically, each cell is assigned with three low-dimensional vectors (in this section, we used dimension 10; different values can be set by the user), one for each kinetic parameter. Similarly, each gene is associated with three low-dimensional vectors of the same dimension, which we term gene effect vectors. The value of each parameter is determined by the dot product of the two respective vectors (Fig. 2a).

The coordinates of a cell's vectors represent factors of cell to cell variability that are extrinsic to the noise generated intrinsically by the process of transcription (which we model by drawing from the stationary distribution above). These values, which we term extrinsic variability factors (EVF) represent a low dimension manifold on which the cells lie and can be interpreted as concentrations of key proteins, morphological properties, microenvironment and more. When simulating a homogeneous population, the EVFs of the cells are drawn from a normal distribution with a fixed mean of 1 and a standard deviation $\sigma$. $\sigma$ is the within-population variability parameter and can be set by the user (for the results in this section $\sigma$ is set to 0.5).

The coordinates of the gene effect vectors can be interpreted as the dependence of its kinetics on the levels of EVFs. For instance, a positive value means that higher concentration of the corresponding EVF can give rise to a higher *on* rate of a certain promoter (if the EVF and gene effect vectors are both for parameter $k_{on}$). The gene effect values are first drawn independently from a standard normal distribution. We then replace each gene effect with a value of zero with probability $\eta$, thus ensuring that every gene is only affected by a small subset of EVFs. The sparseness parameter $\eta$ can be set by the user; in this paper we set it to a fixed value of 0.7.

To map the values of kinetic parameters calculated as dot product of the EVF and gene effect vectors into realistic ranges, we first estimate the distribution of kinetic parameters of genes from real data by fitting a Beta-Poisson model (Methods). To gain a robust estimation of the distribution of kinetic parameters to be used by SymSim, we performed the estimation multiple times with (1) different subpopulations of a dataset; (2) different imputation methods (scVI[4] and MAGIC[18]) to reduce technical variation in real data. Then we obtain aggregated distributions from the results of all the settings we considered (Fig. 2b, Methods, sources of real data described in Data Availability). Notably, the goal of performing kinetic parameters estimation from real data is mainly to identify the range of plausible parameter values to scale the dot products. The ranges in the distributions we obtain (Fig. 2b) are in line with observations from other experiments[19–27] (Supplementary Note 1, Supplementary Table 1). SymSim then applies a quantile approach to map the simulated parameter values resulting from the dot products into the aggregated distributions (Fig. 2a, Methods).

Finally, we account for the possibility of outlier genes with unusually high-expression level, commonly observed in real data. These outlier genes are hard to model with distributional methods, and require additional parametrization[13]. This phenomena is more pronounced in datasets from certain protocols (for example, 10x Chromium[28]) than others (for

example, Smart-seq2[29]), possibly due to selection bias which can be exacerbated by low capture rate. In SymSim, we model the high-expression outlier genes by designating a small subset of genes (whose proportion is determined by the parameter *prop_hge*) as constitutively transcribed, and adjusting their transcription rate $s$ by a factor determined by the parameter *mean_hge* (>1; Methods).

An intriguing question in the analysis of single cell RNA-seq data is the extent to which the conclusion drawn from the data (e.g., stratification into subpopulations) may be confounded by transcriptional bursting and transcriptional noise. SymSim provides a way to explore this. We first note that modality[15,17] and extent of the intrinsic noise[15] in the expression of a gene in a homogenous population of cells (i.e., cells with similar EVFs) can vary for the different ranges of $k_{on}$, $k_{off}$, and $s$. Specifically, one can distinguish the following three types of gene-expression distributions by the number of inflection points in the smoothed density function: unimodal with highest frequency at 0 (no inflection point), unimodal with highest frequency at non-zero value (one inflection point), and bimodal (two inflection points). Figure 2c shows the number of inflection points for different configurations of $k_{on}$ and $k_{off}$ with given $s = 10$. This gives a clear correspondence between kinetic parameter configurations and types of gene-expression distributions. For example, when $s$ is relatively large, we obtain bimodal distributions when $k_{on}$ and $k_{off}$ are smaller than 1.

These results thus guide us in tuning kinetic parameters to obtain desired gene-expression distributions to simulate. Specifically, we focus on adjustment of the bimodality of the distribution, which can lead to large, yet transient fluctuations in mRNA concentration at the same cell over time, thus potentially misleading methods for cell state annotation and differential expression. To increase the overall extent of bimodality in the data, we divide (decrease) all $k_{on}$ and $k_{off}$ values by $10^{bimod}$ (Fig. 2c, yellow arrow). The parameter *bimod* can take value from 0 to 1. This way, other properties such as burst frequency ($k_{on}/(k_{on} + k_{off})$) and synthesis rate ($s$) remain the same. In Fig. 2d we show the effect of varying the *bimod* parameter on gene-expression distribution in a simulated homogenous population. Expectedly, as *bimod* increases, so does the number of bimodal genes, as well as the average Fano factor (Supplementary Fig. 2a).

**Extrinsic variation via extrinsic variability factors**. While the first knob focuses on variation within a homogeneous set of cells, the second knob allows the user to simulate multiple, different cell states. This added complexity is achieved by setting different EVF values for different cells, in a way that allows users to control cellular heterogeneity and generate discrete subpopulations or continuous trajectories. To this end, SymSim represents the desired structure of cell states using a tree (which can be specified by the user), where every subpopulation (in the discrete mode) or every cell (in the continuous mode) is assigned with a position along the tree. Different positions in the tree correspond to different expected EVF values, and the expected absolute difference between the value of an EVF of any two cells is linearly proportional to the square root of their distance in the tree (Supplementary Notes 2 and 7).

When SymSim is applied in a discrete mode, the cells are sampled from the leaves of the tree. The set of cells that are assigned to the same leaf in the tree form a subpopulation, and their EVF values are drawn from the same distribution. As above, we draw these EVF from a normal distribution, where the mean is determined by the position in the tree and the standard deviation is defined by the parameter $\sigma$. When SymSim is applied in a

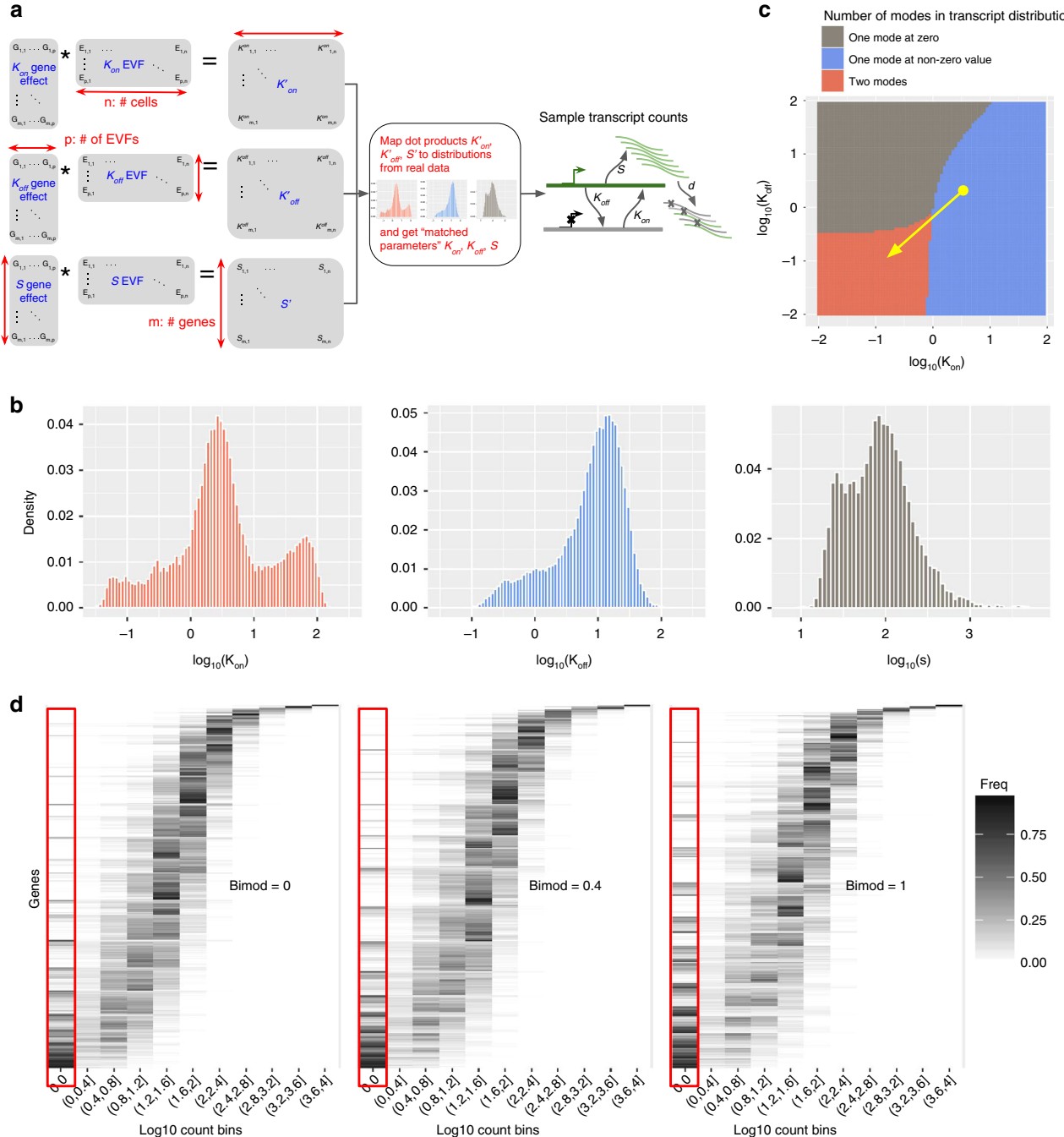

**Fig. 2** Intrinsic variation. **a** A diagram of how gene and cell-specific kinetic parameters are simulated from cell-specific EVF and gene-specific gene effect vectors, and how the kinetic parameters are used in a model of transcription. Each cell has a separate EVF vector for $K_{on}$, $K_{off}$, and $S$. Each parameter is generated through two steps: first, for each gene in each cell, we take the dot product of the corresponding EVF and gene effect vectors. Second, the dot product values are mapped to distributions of parameters estimated from experimental data. The matched parameters are used to generate true transcript counts (see Methods). **b** The distributions of $k_{on}$, $k_{off}$, and $s$ that are used in SymSim for simulations. These distributions are aggregated from inferred results of three subpopulations of the UMI cortex dataset (oligodendrocytes, pyramidal CA1 and pyramidal S1) after imputation by scVI and MAGIC. **c** A heatmap showing the effect of parameter $k_{on}$ and $k_{off}$ on the number of modes in transcript counts. The value of s is fixed to 10 in this plot. The red area with low $k_{on}$ and $k_{off}$ have one zero mode and one non-zero mode. The gray area with low $k_{on}$ and high $k_{off}$ has only one zero mode, and the blue area with high $k_{on}$ and low $k_{off}$ have one non-zero mode. The yellow arrow shows how the parameter *bimod* can modify the amount of bimodality in the transcript count distribution. **d** Histogram heatmaps of transcript count distribution of the true simulated counts with varying values of *bimod*, showing that increasing *bimod* increases the zero-components of transcript counts and the number of bimodal genes. In these heatmaps, each row corresponds to a gene, each column corresponds to a level of expression, and the color intensity is proportional to the number of cells that express the respective gene at the respective expression level. Data used to plot **b–d** can be found in Source Data

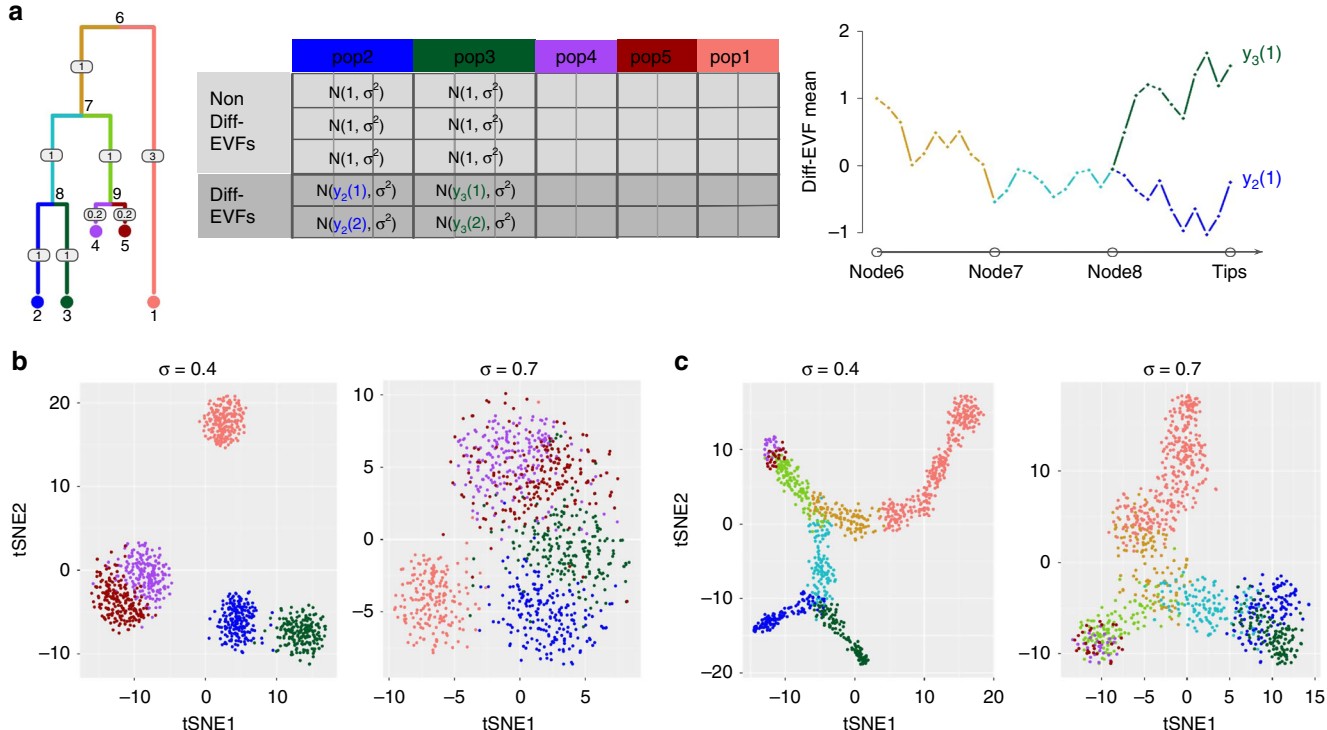

**Fig. 3 Extrinsic variation. a** Illustration of generating a diverse set of cell states with SymSim. The tree represents the relationship between cells. The numbers on the edges are branch lengths; the node numbers indicate the ID of the respective subpopulation (each subpopulation is represented by a single position [leaf] in the tree). The matrix to the right depicts the derivation of EVF values. Each row corresponds to an EVF (only two are Diff-EVF), each column corresponds to a position in the tree, and the content specifies the distribution from which the EVF values are drawn. We use the notation $y_a(b)$ to represent the expected value of EVF $b$ in position $a$ in the tree. The rightmost plot depicts the derivation of these expected values with Brownian motion. We use subpopulations 2 and 3 as examples for both discrete cases (sampling only cells within the subpopulations) or continuous (sampling cells along the trajectories from the root progenitor state [node 6] to the two target subpopulations [nodes 2 and 3]). **b** tSNE plots of five discrete populations generated from the tree structure shown in **a**. Different values of $\sigma$ give rise to different heterogeneity of each population. **c** tSNE plots of continuous populations generated from the same tree. The colors correspond to the colors on branches in the tree shown in **a**. When increasing $\sigma$, cells are more scattered around the main paths which follow the tree structure. Data used to plot **b, c** can be found in Source Data

continuous mode, the cells are positioned along the edges of the tree with a small step size (which is determined by branch lengths and number of cells; Methods). The EVF values are then drawn from a normal distribution where the mean is determined by the position in the tree, and the standard deviation is defined by $\sigma$ (Fig. 3a).

To facilitate the correspondence between EVF values and distances in the tree we use a Brownian motion procedure as described in ref. [30] (Methods; Fig. 3a). Specifically, for each EVF we set the mean value at the root of the tree to a fixed number (default set root node to 1) and then perform Brownian motion along the branch. Fig. 3a illustrates this process using populations 2 and 3 in the tree as an example. Notably, in the continuous mode, this formulation can give rise to a rich set of patterns of changes in gene expression from root (progenitor cells) to leaves (target cells), including the commonly observed impulse profile[31,32] (Supplementary Fig. 3c–d). As an alternative, we also implemented a mode for simulating continuous data by which gene expression from root to leaves is determined explicitly by an impulse function. This might be preferable if the user would like to generate smoother changes in gene expression, or specific temporal patterns. In the following analyses we use the Brownian motion model.

Notably, SymSim only generates a subset of EVFs from the tree, while the remaining ones are drawn from the same distribution for all subpopulations (Fig. 3a). The tree-sampled subset, which we term Diff-EVFs (Differential EVFs) represents

the conditions or factors which are different between subpopulations, and they usually account for a small proportion of all the EVFs. The number of Diff-EVFs can be set by the user. The results in this section were produced with 60 EVFs, 20% of them are Diff-EVFs.

With this formulation, users can control the extent of between-population variation by setting the branch lengths of the input tree, and combine it with a desired level of within-population variation by setting the parameter $\sigma$. Notably, both $\sigma$ and the square root of branch lengths in the tree are in units of EVF values. It is therefore the case that for any two positions in the tree, the ratio of square root tree distance to $\sigma$ determines the separability between the respective distributions of the values assigned to any given Diff-EVF (Supplementary Note 2). As illustration, Fig. 3 depicts the tSNE plots of cells from the same input tree with different $\sigma$ in either a discrete (Fig. 3b) or continuous (Fig. 3c) mode. Notably, both panels show that the tSNE plots reflect the structure of the input tree well.

**Technical variation**. The third knob of SymSim allows users to control technical variation, which accounts for a large part of the variation observed in scRNA-seq datasets[33–35]. The technical confounders reflect noise, reduced sensitivity and bias that are introduced during sample processing steps such as mRNA capture, reverse transcription, PCR amplification, RNA fragmentation, and sequencing. In order to introduce realistic technical variation into our model, we explicitly simulate the major steps in

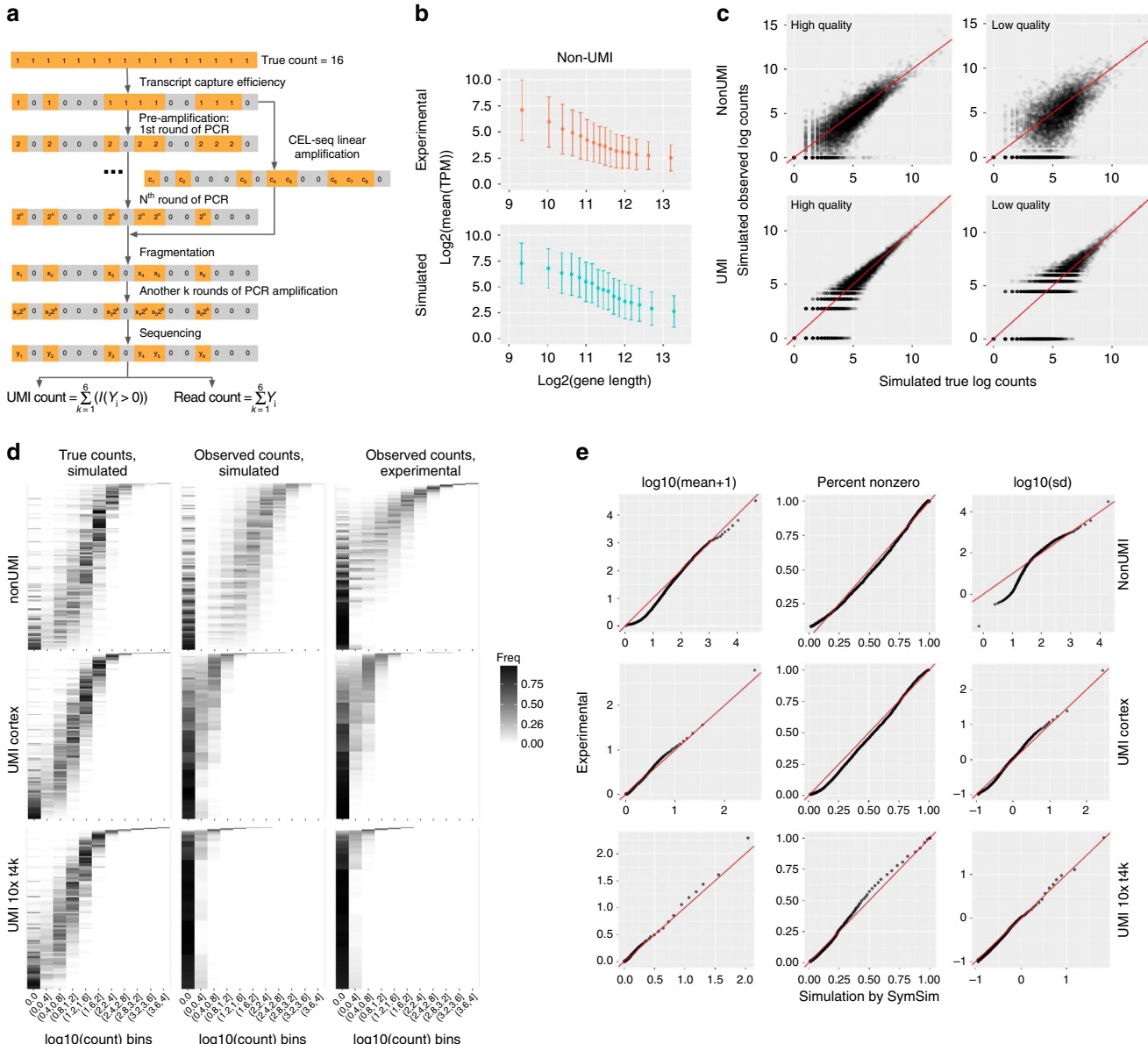

**Fig. 4** Technical variation. **a** A diagram showing the workflow of adding technical variation to true simulated counts. Each gray or orange square represents a molecule of the same transcript in one cell. We implement the following steps: mRNA capturing, pre-amplification (PCR or linear amplification of the cDNAs), fragmentation, amplification after fragmentation, sequencing, and calculation of UMI counts or read counts. Details of these steps can be found in Methods. **b** Gene length bias in both simulated and experimental data for the non-UMI protocol. Error bars represent the ranges of (mean-SD, mean + SD), where SD means standard deviation. **c** Scatter plots comparing true counts and observed counts obtained through: (1) non-UMI, good parameters ($\alpha = 0.2$, $MaxAmpBias = 0.1$, $Depth = 1e6$) for high quality data; (2) UMI, good parameters ($\alpha = 0.2$, $MaxAmpBias = 0.1$, $Depth = 5e5$) for high quality data; (3) non-UMI, bad parameters ($\alpha = 0.05$, $MaxAmpBias = 0.2$, $Depth = 1e6$) for low quality data; (4) UMI, bad parameters ($\alpha = 0.04$, $MaxAmpBias = 0.2$, $Depth = 5e5$) for low quality data. **d** 2D transcript counts histogram heatmap of UMI and non-UMI simulated true counts and simulated observed counts, generated with parameters which best match the input experimental counts, and histogram heatmaps of the respective experimental counts (non-UMI Th17, UMI cortex and UMI 10x t4k datasets). **e** Q–Q plots comparing the mean, percent non-zero and standard deviation in experimental counts and SymSim simulated observed counts respectively for the non-UMI, UMI cortex and UMI 10x t4k datasets. A good match is indicated by most of the dots falling close to the red line. Data used to plot **b–e** can be found in Source Data

the experimental procedures. We implemented two library preparation protocols: (1) full-length mRNAs profiling without the use of UMIs (e.g., with a standard SmartSeq2[29]); and (2) profiling only the end of the mRNA molecule with addition of UMIs (e.g., 10x Chromium[28]). The former protocol is usually applied for a small number of cells and with a large number of reads per cell, providing full information on transcript structure[36]. The latter is normally applied for many cells with shallower sequencing, and it is affected less by amplification and gene length biases[33].

The workflow of these steps is shown in Fig. 4a (Methods). Starting from the simulated true mRNA content of a given cell (namely, number of transcripts per gene, sampled from the stationary distribution of the promoter kinetic model), the first step is mRNA capture, where every molecule is retained with probability $\hat{\alpha}$. The value of the capture efficiency $\hat{\alpha}$ associated with each cell is drawn from a normal distribution with a mean $\alpha$ and standard deviation $\beta$, which can be set by the user. The second step is amplification, where in every cycle SymSim selects each

available molecule with a certain probability and duplicates it. The expected amplification efficiency and the number of PCR cycles can be set by the user. As an optional step, SymSim provides the option of linear amplification (e.g., as in CEL-Seq[37]). We do not apply this option in this manuscript. In the third step each amplified molecule is broken down into fragments, in preparation for further amplification, size selection, and sequencing (Methods).

The number of reads per cell (namely, the number of sequenced fragments) is drawn from a normal distribution whose mean is determined by the parameter *Depth*, which, along with the respective standard deviation (*Depth_sd*) can be provided by the user. To derive the final observed expression values we do not account for sequencing errors, and assume that every sequenced fragment is assigned to the correct gene it originates from. For the non-UMI option, we define the raw measurement of expression as the number of reads per gene. If UMIs are used, SymSim counts every original mRNA molecule only once by collapsing all reads that originated from the same molecule. Notably, for certain depth values, the resulting distribution of number of reads per UMI is similar to the one observed in a dataset of murine cortex cells[38] (Supplementary Fig. 4a, Supplementary Fig. 1G of the cited paper).

It has been previously shown that estimation of gene-expression levels from full-length mRNA sequencing protocols has amplification biases related to sequence-specific properties like gene length and GC-content[33,39], whereas the use of UMIs can correct these biases[39,40]. In particular, we have observed a negative correlation between gene length and length-normalized gene-expression in our reference non-UMI dataset (murine Th17 cells from Gaublomme et al.[41]; Fig. 4b), and the same trend is reported by Phipson et al.[39]. To account for that, we parametrize the efficiency of the PCR amplification step using a linear model that represents gene length bias (Methods). As a result, our simulated data with a non-UMI protocol show a similar dependence of gene-expression on gene length as in experimental data (Fig. 4b, real data is from ref. [41]). In cases where UMIs are used, gene length effects are also modeled during amplification, but these effects are mitigated since each molecule is counted at most once. We therefore do not observe gene length bias in the UMI-based simulated data, similarly to the experimental data (Supplementary Fig. 4b, real data is from ref. [38]). Finally, we model batch effects with multiplicative factors that are gene- and batch-specific. In Supplementary Fig. 4c, we show the same population of cells are separated by batches. To simplify the discussion at the remainder of this paper, we assume that the data come from a single batch.

In Fig. 4c, we show the comparison between the simulated true mRNA content of one cell and the simulated observed counts obtained with or without UMI. We consider two scenarios: the first scenario represents a study with a low technical confounding and the second one represents a highly confounded dataset. Parameters which differ between these "good" and "bad" cases in this example include capture efficiency (*α*), extent of amplification bias (*MaxAmpBias*), and sequencing depth (*Depth*). Using "bad" technical parameters introduce more noise to true counts, and compared to the non-UMI simulation the UMIs reduce technical noise. The histograms of true counts and four versions of simulated counts are shown in Supplementary Fig. 4d. Using quantile-quantile plots (Q–Q plots; Supplementary Fig. 4e) further demonstrates that UMIs help in maintaining a better representation of the true counts in the observed data.

The total computation time to simulate a dataset consists of time to generate true counts and time to generate observed counts from true counts. We show the runtime and memory usage for different parameter configurations in Methods.

**Fitting parameters to real data**. For a given real dataset, SymSim can produce observed (read or UMI) counts that have similar statistical properties to the real data (Fig. 4d–e), by searching in a database of simulations obtained from a range of parameter configurations (Methods). This procedure focuses on within-population variability (similarly to Splatter[13]) and sets the values of ten parameters from both the first and third knobs (Methods). We test this function with the non-UMI Th17 dataset[41] (using all cells), the UMI cortex dataset[38] (using a subpopulation of 948 CA1 pyramidal neuron cells) and two UMI datasets from 10x Genomics, denoted by "UMI 10x t4k" and "UMI 10x pbmc8k" (details and sources of experimental data are described in Data Availability). See Supplementary Note 4 and Supplementary Tables 2–5 for the values of the fitted parameters.

Side by side inspection of the histograms of true mRNA levels (simulated) and observed counts (simulated and experimental), indicates that SymSim can transform the simulated ground truth (Fig. 4d, left) into simulated observations (Fig. 4d, middle) that match the real observed data for both UMI and non-UMI protocols (Fig. 4d, right). For a more quantitative analysis, we generated Q–Q plots of the distributions of mean (after adding one to all values), percent non-zero and standard deviation (SD) of genes between simulated and experimental data (Fig. 4e, Supplementary Fig. 5b for the UMI 10x pbmc8k dataset). Notably, we observe a certain level of inaccuracy in matching the SD at the lower ends for the non-UMI data, which can be due to lowly expressed genes. Indeed, when we exclude lowly expressed genes from the real data, the matching of SD improve substantially (Q–Q plots shown in Supplementary Fig. 5a). Furthermore, we conducted a similar analysis by training Splatter[13] and powSimR[12] with the same experimental datasets as input, and found that SymSim matches this data significantly better (Supplementary Fig. 5c). Q–Q plots of the distributions of coefficient of variation (CV) and mean (without adding one) of genes between experimental data and data simulated, respectively, by SymSim, Splatter and powSimR also show that SymSim provides an overall better fit than the other two simulators (Supplementary Fig. 5d–e). We also performed additional comparisons of the simulators with other measurements, described in Supplementary Note 5 and shown in Supplementary Figs. 6a–b and 7a–c.

**Comparing computational methods for single cell RNA-seq data**. SymSim can be used to benchmark methods for single cell RNA-Seq data analysis as it provides both observed counts and a reference ground truth. In the following sections we demonstrate the utility of SymSim as tool for benchmarking methods for clustering, differential expression, and trajectory inference in a sample consisting of multiple subpopulations, using the structure depicted in Fig. 3a. The design of SymSim allows us to evaluate the effect of various biological and technical confounders on the accuracy of downstream analysis. Here, we investigate the effect of total number of cells (*N*), within population variability (*σ*), mRNA capture rate (*α*), and sequencing depth (*Depth*). We also test the effect of the proportion of cells associated with the smallest subpopulation of cells (*Prop*), using population 2 in the tree as our designated rare subpopulation.

**Using SymSim to evaluate clustering methods**. We begin by inspecting the impact of each parameter on the performance of clustering methods. To this end, we simulated observed counts using the UMI option, and traversed a grid of values for the five parameters with 18 simulation runs per configuration. The values of the remaining parameters are largely determined according to the cortex dataset[38] and specified in Supplementary Table 6. We

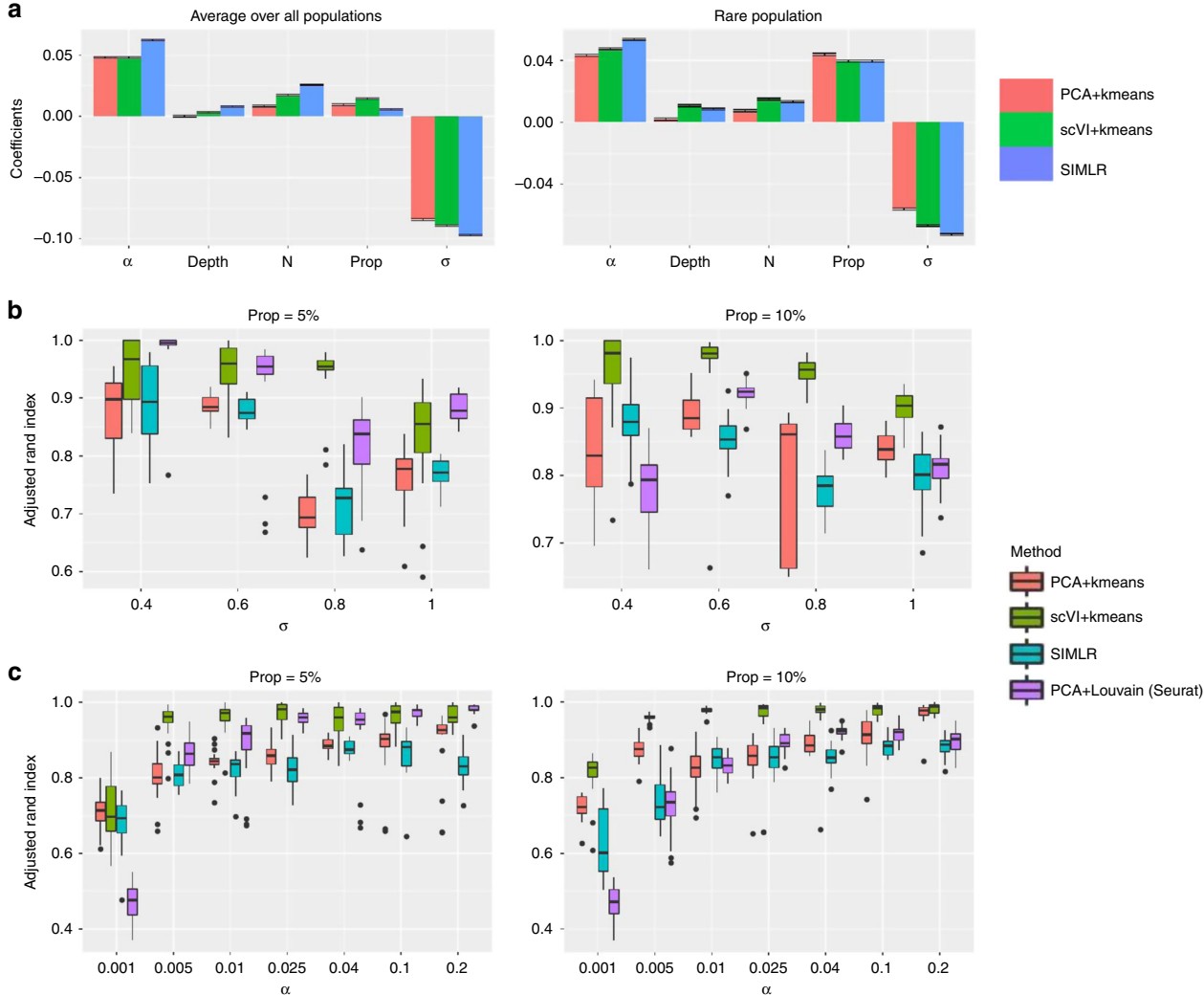

**Fig. 5** Benchmarking of clustering methods. **a** Coefficients of various parameters from multiple linear regression between parameters and the adjusted Rand index (ARI). In the left plot the ARI are averaged over all populations, and in the right plot the ARI is only for the rare population (population 2). **b** ARI of the rare populations using the four clustering methods when changing $\sigma$ ($\alpha = 0.04$). Left plot: the rare population accounts for 5% of all the cells; right plot: the rare population accounts for 10% of all the cells. **c** ARI of the rare populations using the four clustering methods when changing $\alpha$ ($\sigma = 0.6$). Left plot: the rare population accounts for 5% of all the cells; right plot: the rare population accounts for 10% of all the cells. Data used to plot **a**–**c** can be found in Source Data

tested three clustering methods: k-means based on Euclidean distance of the first 10 principle components, k-means based on Euclidean distance in a nonlinear latent space learned by scVI[4] and SIMLR[42]. In all cases we set the expected number of clusters to the ground truth value ($k = 5$). The accuracy of the methods is evaluated using the adjusted Rand index (ARI; higher values indicate better performance). To inspect the effects of the various parameters on clustering performance, we performed multiple linear regression between the parameters and the ARI. The regression coefficients are shown in Fig. 5a. Overall, $\sigma$ appears to be the most dominant factor, and the proportion of the rare population (*Prop*) is clearly positively associated with better performance. Among the technical parameters, while $\alpha$ plays a role on the performance especially for the rare population, the impact of *Depth* is minor.

We then focus on varying the dominant factors (except *N*, which we discuss in the next section), namely, the within-population variation parameter $\sigma$ and mRNA capture efficiency $\alpha$, for the benchmarking analysis henceforth. In particular, the range

of values for $\sigma$ is set such that we cover various data characteristics, from well separated populations, to almost entirely mixed ones (Supplementary Table 6). We compare the performance of four clustering strategies: SIMLR[42], dimensionality reduction with scVI[4] followed by k-means, and dimensionality reduction with PCA followed by Louvain clustering[43] (implemented in Seurat[44]) or k-means. We observe better accuracy as the quality of the data increases or the within-population variation decreases (Fig. 5b–c). Interestingly, comparing $\sigma = 0.6$, $\sigma = 0.8$, and $\sigma = 1$, we can tell that when $\sigma$ is high enough to make the clustering challenging, further increasing $\sigma$ does not yield obvious changes (Fig. 5b). We observe a similar trend of saturation, inspecting increasing levels of capture efficiency ($\alpha$), especially with scVI. Comparing the methods to each other, we see that scVI/k-means has the highest ARI in most cases; when the rare population accounts for 5% of all cells, Seurat is the second best and PCA/k-means and SIMLR are comparable; when the rare population accounts for 10% of all cells, SIMLR performs slightly worse than Seurat and PCA/k-means.

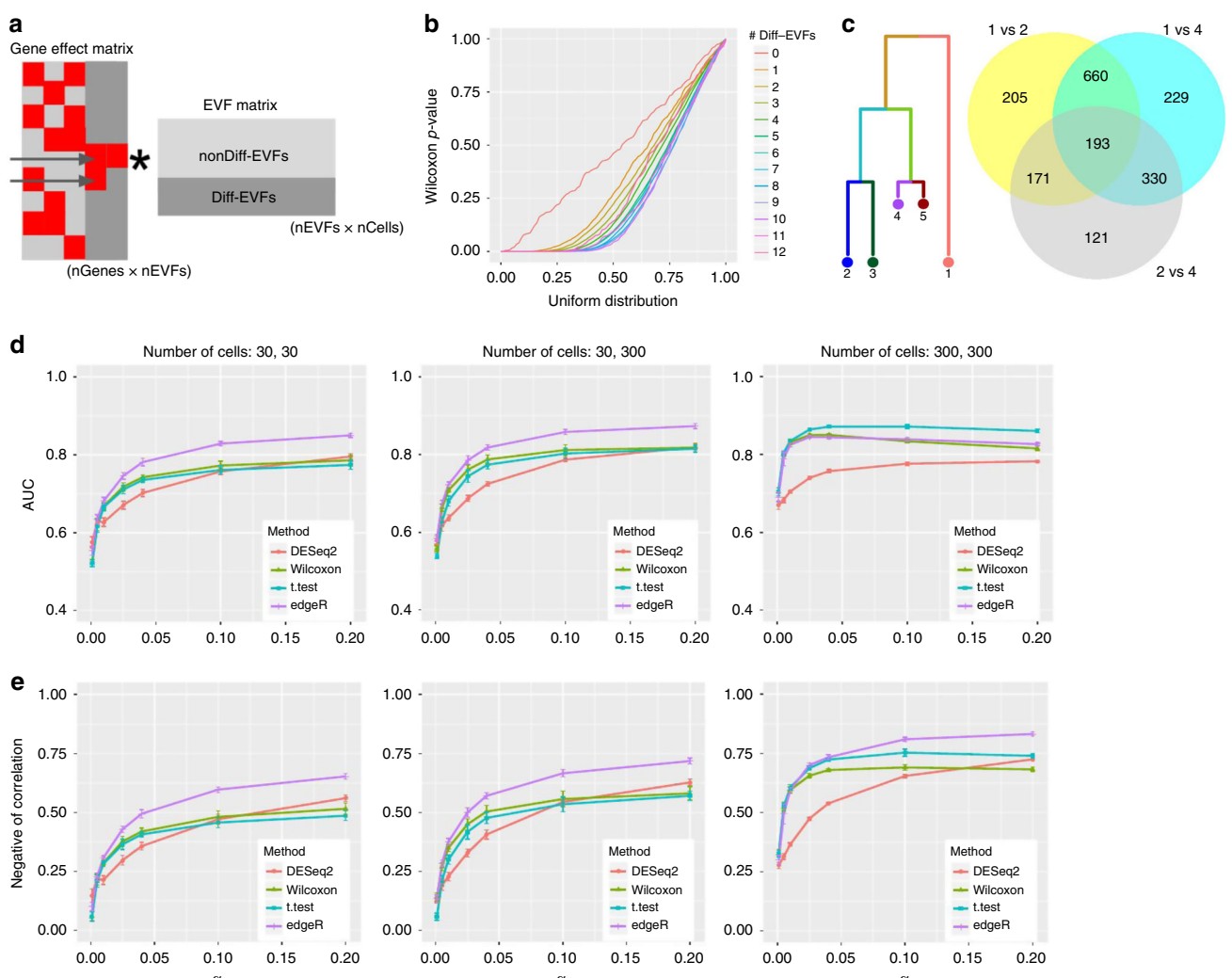

**Fig. 6** Benchmarking of DE detection methods. **a** Illustration of how DE genes are generated through the Diff-EVFs. Red squares in the gene effect matrix correspond to non-zero values. The two genes indicated by the arrows are DE genes by number of Diff-EVFs they have (respectively, 2 and 1). **b** Q–Q plot comparing the p-value obtained from differential expression analysis between subpopulations 2 and 4 (using Wilcoxon test on the true simulated counts) to a uniform distribution. Genes are grouped by the number of Diff-EVFs they use and different groups are plotted in different colors. The numbers of Diff-EVFs used by genes can be thought of as the degree of DE-ness. Genes with more Diff-EVFs have p-values further diverged from uniform distribution. **c** Venn diagram showing that closely related populations have less DE genes between them compared to distantly related populations. We use populations 1, 2, 4 as examples: there are much more DE genes from comparison of "1 vs 2" and "1 vs 4" than "2 vs 4", and DE genes from "1 vs 2" and "1 vs 4" have a big overlap. The DE genes are determined by log2 fold change (LFC) of true counts with criterion |LFC| > 0.8. **d** The AUROC (area under receiver operating characteristic curve) of detecting DE genes using four different methods from observed counts with changing capture efficiency $\alpha$ ($\sigma = 0.6$). The populations under comparison are 2 and 4. Three sets of criteria were used to define the true DE genes and the final performance was the average performance from the three sets: (1) nDiff-EVFgene > 0 and |LFC| > 0.6; (2) nDiff-EVFgene > 0 and |LFC| > 0.8; (3) nDiff-EVFgene > 0 and |LFC| > 1. LFC was calculated with theoretical means from the kinetic parameters. **e** The negative of correlation between log2 fold change on theoretical mean of gene-expression and p-values obtained by a DE detection method, with changing capture efficiency $\alpha$ ($\sigma = 0.6$). The populations under comparison are 2 and 4. Data used to plot **b**, **d**, **e** can be found in Source Data

**Using SymSim to evaluate differential expression methods.** Our mechanism for simulating multiple populations automatically generates differentially expressed (DE) genes between populations (in the discrete setting; Fig. 3b) or along pseudotime (in the continuous setting; Fig. 3c). In the following, we use SymSim to benchmark methods for detecting DE genes, focusing on the discrete setting. We use two criteria to define the ground truth set of DE genes. The first criterion is that the number of Diff-EVFs that are associated with a non-zero gene effect value (which we denote as nDiff-EVFgene; Fig. 6a) should be larger than zero. This criterion is motivated by our model of transcription regulation: the kinetic parameters of a gene are affected by extrinsic factors, and changes to extrinsic factors might

therefore lead to changes in the number of transcripts. Indeed, when we compare the true simulated gene expression values between subpopulations (i.e., before introducing technical confounders), we get a uniform (random) distribution of p-values for genes with no Diff-EVFs, and an increasing skew as nDiff-EVFgene increases (Fig. 6b, using Wilcoxon test); Supplementary Fig. 9a shows that the log fold change of gene-expression between subpopulations increases with nDiff-EVFgene. An additional constraint for a gene being differentially expressed is that it must have a sufficiently large fold change in their simulated true simulated expression levels (threshold of absolute log2 fold change ranges from 0.6 to 1, details in Figure legends; Methods).

An important distinguishing feature of SymSim is that it provides an intuitive way for generating case studies for DE analysis that consist of multiple subpopulations with a predefined structure of similarity. To illustrate this, consider populations 1, 2, and 4 (Fig. 6c), which form a hierarchy (2 and 4 are closer to each other and similarly distant from 1). This user-defined structure is reflected in the sizes of the sets of DE genes, obtained, respectively, from populations 1 vs 2 (1229 genes), 1 vs 4 (1412 genes), and 2 vs 4 (815 genes). Consistent with the hierarchy, the first two gene sets are overlapping and larger than the third one.

As an example for a benchmark study, we used four methods to detect DE genes: edgeR[45], DESeq2[46], Wilcoxon rank-sum test, and $t$-test on observed counts generated by various parameter settings (Methods, Supplementary Table 7). We tested the effect of the total number of cells ($N$) and mRNA capture rate ($\alpha$) with 10 simulation runs per parameter configuration. We use two accuracy measures: (a) AUROC (area under receiver operating characteristic curve), obtained by treating the p-values output from each method as a predictor (Fig. 6d, Methods); (b) negative of Spearman correlation between the p-values of each detection method and the log fold difference of the true expression levels (Fig. 6e, Methods).

From Fig. 6d–e, one can observe that when the numbers of cells are small (30 in each population), edgeR has the best performance while the other three methods are comparable to each other. When the numbers of cells of both populations increase to 300, the two naive methods Wilcoxon test and $t$-test improve in their relative performance, compared to edgeR and DESeq2. The case where the numbers of cells are 30 and 300 appears to have performance between those of the 30 vs 30 case and the 300 vs 300 case. When increasing capture efficiency, all methods gain performance except for the case of AUROC with 300 cells. In that case, the drop in AUROC for some methods is caused by inflation in p-values as $\alpha$ increases, which results in lower specificity (Supplementary Fig. 9b). Notably, we noticed that the adjusted p-values from DESeq2 can have many missing entries (NAs), especially when $\alpha$ is low (and thus counts are low), and therefore we used its unadjusted p-values in Fig. 6d–e. However, this assignment of NAs in practice filters out genes, which do not pass a certain threshold of absolute magnitude (explained in DESeq2 vignette[47]). To make use of this filtering, we conducted an additional analysis where we used the adjusted p-values for DESeq2 and compare it to all other methods using only the non-filtered (non NA) genes (Supplementary Fig. 9c). As expected, the performance of all methods (and specifically DESeq2) improves when considering only this set of genes, and converges to high values already at lower capture efficiency rates.

To summarize, we find that edgeR has the best overall performance, with the $t$-test rank second followed by Wilcoxon test. This ranking is consistent with results from a recent paper which evaluated 36 methods for DE analysis with single cell RNA-Seq data[48].

We also investigate the effects of bimodality (controlled by parameter *bimod*) on the performance of clustering and differential expression algorithms. This analysis is presented in Supplementary Note 6 and Supplementary Figs. 8a–b and 10a–d.

**Using SymSim to evaluate trajectory inference methods**. The ability of SymSim to generate a continuum of cell states makes it a convenient choice to benchmark trajectory inference methods. We compare three methods including Monocle[49,50], Slingshot[51], and a minimum spanning tree (MST) algorithm implemented in the package dynverse[52] (Methods). We generate datasets with different values of $\sigma$ and $\alpha$ with the input tree shown in Fig. 3a. For each parameter configuration, we repeat the simulation 10

times. To evaluate the trajectory inference methods, we use two measures: (1) Spearman correlation between true cell order and inferred cell order. We consider cells on each lineage (a path from root to a leaf) separately and take the average of correlation on all five lineages. (2) $k$-nearest neighbor purity (knn purity) of cells, that is, for each cell, we calculate the Jaccard Index between its $k$-nearest neighbors in the true trajectory and that in the inferred trajectory. Results are shown in Fig. 7. In these plots, $k$ is set to 100. In Fig. 7a, we vary $\sigma$ and fix $\alpha$ as 0.1. Both the correlation and knn purity decrease when $\sigma$ increases. In Fig. 7b, we vary $\alpha$ and fix $\sigma$ as 0.6. All methods show an overall increasing trend along with $\alpha$ with both measures. Consistent with a recent benchmark study[52], we observe that overall Slingshot clearly outperforms the other two methods.

**Experimental design**. Deciding how many cells to sequence is a decision many researchers face when designing an experiment, and the optimal number of cells to sequence highly depends on the nature of the biological system under investigation and the respective technical hurdles. A previous approach to this problem[53] assumes that the goal of the experiment is to identify subpopulations of cells and provides a theoretical lower bound for the problem. This bound considers the aspect of counting cells (namely, sequencing enough representative cells from each subpopulation), but it does not account for the identifiability of each subpopulation, which may be hampered by both technical and biological factors as well as the performance of clustering algorithms.

In the following we demonstrate how SymSim can be used to shed more light on this important problem. Importantly, in its current form SymSim does not use real data to model between-population variability. We therefore interpret the results in a relative manner—how do different variability factors shift the required number of cells, compared to each other and to the theoretical lower bound. Our example focuses on a case of one rare subset, represented by cells from population 2 (using the same tree in Fig. 6c; note that one can easily generalize this procedure to multiple rare subpopulations). We simulate observed counts with numbers of cells ($N$) ranging from 600 to 7000. These simulations were based on the parameters fit to the cortex dataset[38] with varying levels of $\sigma$ and $\alpha$ (250 simulations per parameter configuration).

We applied the same four clustering methods as described in the previous section ($k$-means with scVI or PCA, Louvain clustering with PCA (Seurat) and SIMLR). We say that a given algorithm was successful in detecting the rare population if at least 50 cells from this set are assigned to the same cluster, and form at least 70% of the cells in that cluster. We use these labels to compute an empirical success probability $P$ for each algorithm and each parameter configuration. Out of the 250 simulations for each parameter configuration, we randomly sample 100 randomizations 20 times, and for each 100 randomizations we can calculate a value of $P$. We then plot the mean and standard deviation as error bars of the 20 values of $P$ for each $N$ under each configuration (Fig. 8a–d, Supplementary Fig. 11). To get an upper bound on performance that better reflects the data (rather than the choice of algorithm), we take the best $P$ out of all algorithms, and apply cubic spline smoothing (gray curves, Fig. 8a–d). In each plot we also include the theoretical limit which only requires the presence of at least 50 cells from the rare subpopulation (Methods). The theoretical curve (which is independent of all parameters except $N$) reaches almost 1 at $N = 1400$. Conversely, the empirical curves vary dramatically, based on parameter values. For an easy case of low within-population variability ($\sigma = 0.4$) and high capture efficiency ($\alpha = 0.1$) the empirical upper

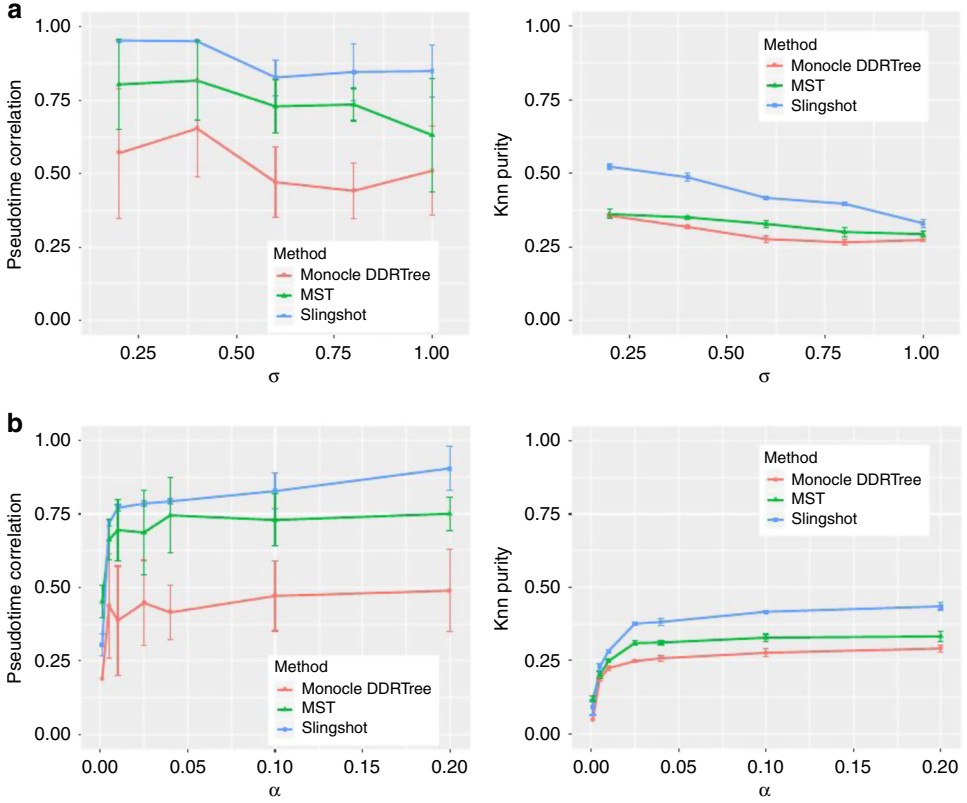

**Fig. 7** Benchmark trajectory inference methods. **a** Pseudotime correlation and knn purity of all methods when varying $\sigma$ ($\alpha = 0.1$). **b** Pseudotime correlation and knn purity of all methods when varying $\alpha$ ($\sigma = 0.6$). Data used to plot **a**, **b** can be found in Source Data

bound curve is close to the theoretical one (Fig. 8a). This curve decreases when increasing the effect of either nuisance factor (Fig. 8b–c). The reduction is substantially more dramatic for most of the methods when both nuisance factors increase, while Seurat remains robust to this change, potentially due to that the graph-based clustering method is advantageous in reducing false positives for the rare population compared to k-means (Fig. 8d).

To understand the implications on the number of cells required in a given setting, we calculated how many cells are required, in each configuration, to achieve a success rate of respectively 0.6, 0.7, 0.8, and 0.9 (Fig. 8e). As expected, the resulting numbers can be much higher than the theoretical lower bound. For example, to achieve a success rate of 0.9, when the within-population variability increases ($\sigma = 0.7$), we need at least 3838 cells (corresponding to $\alpha = 0.01$), while with the theoretical curve, we need only less than 1200 cells. In general, the number of cells needed increase when the desired success rate and $\sigma$ increase. Very low capture efficiency ($\alpha = 0.001$, $\alpha = 0.005$) tend to require high number of cells. Considering only the binomial sampling of cells may therefore underestimate the number of cells needed for a realistic scenario, and considerations of biological and technical variations with simulators like SymSim is merited.

## Discussion
SymSim has the following features, which are advantageous over existing simulators: (i) We simulate true transcript counts from a kinetic model that can be interpreted in terms of transcript synthesis rate, promoter activation, and deactivation. (ii) When generating multiple discrete or continuous populations, instead of generating biological differences through directly altering the true transcript count distribution, we set Diff-EVFs, which can be interpreted as biological conditions that cause the differences

between subpopulations of cells. This is a more natural and realistic way to simulate biological transcriptional differences. (iii) The EVF formulation provides an intuitive way to specify and simulate complex structures of cell–cell similarity, without the need for manual specifications of the numbers of DE genes[13]. (iv) When generating observed counts, we simulate key steps in real experimental protocols, which automatically gives us dropout events, length bias, and distribution of library sizes. We also provide choices to use UMI-based protocols or non-UMI full-length mRNA protocols, as the properties of data output from these two categories can be very different.

The main input parameters to SymSim, mostly the parameters in the third knob, are self-explanatory with their own technical meanings, which users can adjust to match an experimental dataset of interest. SymSim allows users to simulate datasets with desired properties or matched with experimental data. While the procedure of parameter fitting was developed in order to generate simulated datasets with similar properties, it may also provide additional insight, as the parameters are biologically or technically interpretable. For instance, comparing the parameters fit to the the UMI and non-UMI datasets in this study we note that the capture efficiency inferred to the latter is much higher (Supplementary Tables 2–5). The modular nature of SymSim provides possibilities to generalize its application. For example, the generation of true counts with EVFs and transcription kinetics can be replaced by learning a generative model from real data, with methods such as scVI[4]. This type of extension will facilitate simulation of between-subpopulation diversity that better mimics experimental observations, albeit at the cost of using parameters that are less interpretable biologically. Another extended application of interest is to use different tree structures for different Diff-EVFs when generating multiple populations of cells, such

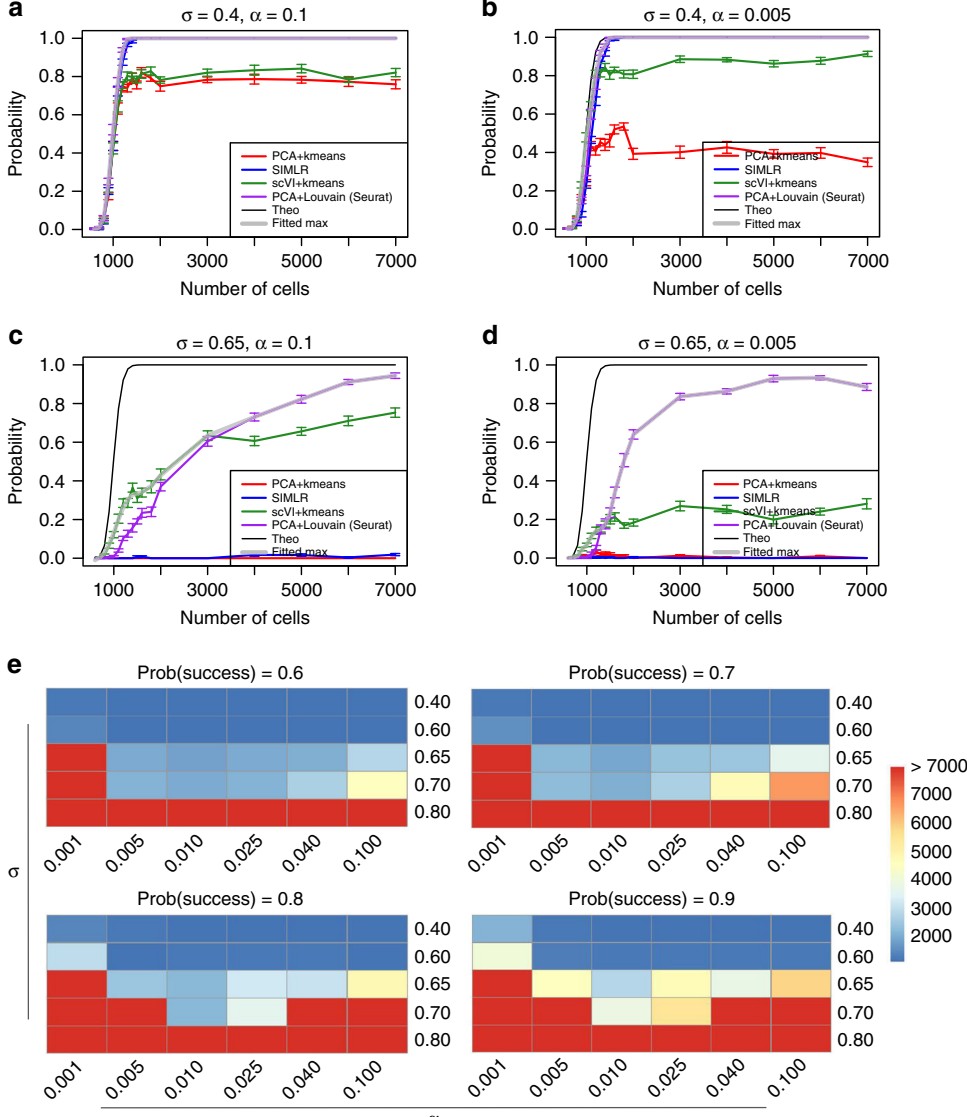

**Fig. 8** The number of cells needed to detect a rare population. We generate five populations according to the tree structure shown in Fig. 3 and set population 2 as the rare population which accounts for 5% of the cells. Other populations share 95% of the cells evenly. The criteria of detecting the rare population are that at least 50 cells from this population are correctly detected and the precision (positive predicted value) is at least 70%. **a–d** The probability of detecting the rare population when sequencing N (x-axis) cells under different σ and α configurations, with different clustering methods. The black curve represents the theoretical probability from the binomial model, assuming that all cells sequenced are assigned correctly to the original population. The gray curve with transparency takes the maximum value at each data point from all four clustering methods with smoothing. Error bars are standard deviation over 20 randomizations. **e** The heatmaps show the number of cells needed to sequence under different configurations of σ and α to detect the rare population with success rates 0.6, 0.7, 0.8, 0.9, always using the best clustering method. Data used to plot **a–e** can be found in Source Data

that every tree represents a different aspect of variability between cells. For instance, using this approach, one tree can represent a differentiation process and the other can represent variability due to the physical location of the cell.

As the number and extent of biological applications of single cell genomics continues to grow, so does the extent of analytical questions one can tackle, which go beyond standard bulk era analysis steps (e.g., trajectory analysis, mRNA velocity[54], and more). The need for robust analytical methods therefore increases, and so does the means for proper evaluation of these methods. SymSim provides a starting point to address this challenge of flexible and feature-rich simulation for method evaluation, as it aims to directly mimic the key mechanistic properties of single cell RNA sequencing.

## Methods

**Simulating gene expression with the kinetic model**. As shown in Fig. 2a, the kinetic model of gene expression considers that a gene can be either *on* or *off* and the probabilities to transit between the two states are $k_{on}$ and $k_{off}$. When the gene is *on* it is transcribed with transcription rate *s*. The transcripts degrade with rate *d*. For a given gene, based on these parameters one can simulate the number of its transcript molecules over time. The theoretical probability distribution can be calculated via the Master Equation[15,17], which is the steady state solution for the kinetic model. Alternatively, the kinetic model can be represented by a Beta-Poisson model[17], which we use in our implementation to sample expression values for a gene.

**Calculating parameters for the kinetic model in SymSim**. For a gene in a cell, the parameters for the kinetic model $k_{on}$, $k_{off}$, and *s* are calculated from the cell-specific EVF vectors of this cell and the gene effect vectors of the gene (Fig. 2a). To allow independent control of the three parameters, we use one EVF vector and one

gene effect vector for each parameter. Take $k_{on}$ as an example: denoting the EVF vector as $(e_1^{kon}, e_2^{kon}, \ldots, e_p^{kon})$, and the gene effect vector for $k_{on}$ as $(g_1^{kon}, g_2^{kon}, \ldots, g_p^{kon})$, the cell-gene specific value for $k_{on}$ is the dot product of these two vectors. We then map these $k_{on}$ values to the distribution of $k_{on}$ estimated from experimental data, to obtain the matched parameters. We do so by sorting the $k_{on}$ values (from dot products) for all genes in all cells, sampling the same number of values from the experimental $k_{on}$ distribution (the number of values would be $m*n$, where $m$ is the number of genes and $n$ is the number of cells), and updating the $k_{on}$ values to the ones sampled from the experimental distribution with the same rank. The values of $k_{off}$ and $s$ are calculated in the same way.

Finally, to better model the high-expression outlier genes which usually account for a very small proportion of all genes, we introduce two parameters (*prop_hge* and *mean_hge*) to SymSim for these outlier genes. First, we use parameter *prop_hge* to represent the proportion of these genes. For randomly selected genes according to this proportion, we consider that they are always on and their gene-expression levels are sampled from a Poisson distribution with parameter $s$ (the transcription rate). To reflect their high gene-expression level, we increase their transcription rate $s$ calculated from EVFs and gene effects with an inflation factor. The inflation factor for each gene is the exponential (base 2) of ($mean\_hge-1 + rank/n$), where *rank* is the rank of mean $s$ of a gene among all outlier genes (smaller $s$ corresponds to smaller rank values), $n$ is the total number of outlier genes and *mean_hge* specifies the extent of inflation to original $s$. The values of *prop_hge* and *mean_hge* can be fitted from real data.

**Estimating kinetic parameters from real data**. We estimated kinetic parameters from experimental data using an MCMC approach. For each gene, its expression $X$ depends on $p$, the proportion of time it is *on*, and the mRNA synthesis rate $s$. The parameter $p$ itself is a random variable determined by the kinetic parameters $k_{on}$ and $k_{off}$. We model $p$ as a Beta distributed variable with shape parameters $k_{on}$ and $k_{off}$. We model X as a Poisson-distributed variable with parameter $p*s$. The distribution of X is then identical to the distribution calculated using the Master Equation[17]. The downsampling effect is modeled as a Binomial sampling with X being the number of trials, and $f$ being the probability that a transcript is sampled for sequencing.

We fit this model to the experimental data using the Gibbs sampler implemented in RJAGS. The number of iterations is set to 2000. At every iteration, we sample each parameter from its marginal posterior conditional on the value of all other parameters. To meet the assumption that all cells share the same kinetic parameters we divide cells by clustering that is performed in the original study and fit the model to counts in a single cluster of cells at a time. We use imputed read counts, rather than the raw read counts. We use scVI[4] and MAGIC[18] for the imputation. We use multiple starting points to diagnose convergence. For each cell cluster and each imputation method, we fit the model independently three times. We observe that different runs from the same data have very similar distributions (Supplementary Fig. 2c, UMI cortex data, subpopulation pyramidal CA1, imputed with scVI). We also investigate the autocorrelation and plotted the typical lag vs correlation plots for these chains (Supplementary Fig. 2d). The correlation drops as the lag increases, which is a sign of sufficient mixing of samples. We then merge all the acceptable chains to obtain the distributions for the combination of cluster and imputation method. Finally, results from all combinations are merged to obtain the final distribution of kinetic parameters (Fig. 2b).

As our reference, we used a UMI-based dataset of 3005 cortex cells by Zeisel et al.[38] and a non-UMI-based dataset of 130 IL17-expressing T helper cells (Th17) by Gaublomme et al.[41] (See Data Availability for further details on the experimental data). To mitigate the effects of low sensitivity, the UMI-based data were imputed using scVI[4] and MAGIC[18], and the non-UMI data were imputed using MAGIC[18] (as scVI is only applicable to large datasets). To reduce the effects of extrinsic variation, we performed the parameter estimation separately in each of the three largest clusters in the cortex dataset (each cluster is assumed to represent a relatively homogeneous subpopulation), and on the entire T cell data (a single condition, which did not contain obvious clusters) and obtained similar distribution ranges (Supplementary Fig. 2b).

We then perform simulations to test how well this procedure of kinetic parameter estimation reconstructs the true parameters. We simulate observed UMI counts for five discrete populations whose relationships are defined by the tree in Fig. 3a, with the complete workflow of SymSim. Then we perform imputation on the observed UMI counts with both scVI and MAGIC. We then apply the kinetic parameter estimation procedure described above on each subpopulation imputed by each imputation method separately. Finally we aggregate the parameters estimated from all populations and both imputation methods, and gain the distributions estimated from imputed counts. We also apply the kinetic parameter estimation procedure on the true counts which are available (Supplementary Fig. 3b).

**Simulation of discrete and continuous populations**. The structure of populations can be represented by a tree and the user can input the tree in Newick format in a text file. The differences between populations are realized through Diff-EVFs, which usually account for a small proportion of all EVFs. There are two different modes of simulation the Diff-EVFs, Continuous and Discrete. Both modes can be

modeled by Brownian motion along the tree from root to leaves, where one starts with a given value at the root (default is 1), and at each time point $t$, $y(t)$ is calculated as $y(t) = y(t-1) + N(0, \Delta t)$, where $N()$ represents a Gaussian function, and $\Delta t$ is the step size. The values at internal nodes of the tree are shared by all branches connecting this node. In the continuous mode, the step size between two consecutive cells on a given branch is obtained by randomly sampling $n_b$ positions on a branch $b$ of length $l_b$. In the discrete mode, the step size is the corresponding branch length and the number steps is the depth of the tips. For a given tip and a given EVF, the value we sample at the tip is used as the mean of a Gaussian distribution to sample the values for that EVF for all cells in that population with standard deviation $\sigma$ (Fig. 3a).

For the continuous mode we also provide an alternative option to the Brownian motion model: using impulse functions for modeling the path-specific variation. When impulse function is used, for cells sampled from branches that are not on the root-tip path for the specific EVF, they are sampled from a univariate normal with mean equal to the EVF value at their most recent common ancestor with the varying path, and standard deviation $\sigma$.

**Simulating technical steps from mRNA capturing to sequencing**. We simulate two categories of library preparation protocols, one does not use UMIs (unique molecular identifiers)[28] and sequences full-length mRNAs (using procedures in Smart-seq2[29] as template), and the other uses UMIs and sequences only the 3′ end of the mRNA (using the Chromium chemistry by 10x Genomics as template). In the pre-amplification step, we provide option of using linear amplification to mimic the CEL-seq protocol. As shown in Fig. 4a, we take one transcript with 16 molecules as an example. To implement the UMIs, each original molecule has a variable to its count at each step. The technical steps include the following:

(1) Capturing step: molecules are captured from the cell with probability $\alpha$.

(2) Pre-amplification step: if using non-CEL-Seq protocols, this step involves N rounds of PCR amplifications. We introduce sequence-specific biases during amplification, which includes transcript length bias and other bias assigned randomly. Parameter *lenslope* can be used to control the amount of length bias, and *MaxAmpBias* is used to tune the total amount of amplification bias. If using CEL-Seq protocol this step is in vitro transcription (IVT) linear amplification.

(3) Fragmentation step: the mRNAs are chopped into fragments for sequencing. If sequencing full-length mRNA, all fragments with acceptable length are kept for sequencing. If sequencing only the 3′ end for UMI protocols, only fragments on the 3′ end are kept for sequencing. The lengths of the simulated transcripts are obtained from the human reference genome, and the fragmentation is calibrated so that the average fragment length is 400 bp, which is typical for RNA sequencing (Supplementary Note 3). For each transcript length, we calculate a distribution of number of fragments given expected fragment length, and use this distribution to generate the number of fragments during our simulation of the fragmentation step. The distributions are different for non-UMI and UMI protocols; the details of calculating the distributions are in Supplementary Note 3. Resulting fragments that are within an acceptable size range (100–1000 bp) are then carried on to the next step.

(4) Amplification step: fragments go through another k rounds of PCR amplifications for all protocols, including CEL-Seq and non-CEL-Seq protocols.

(5) Sequencing step: amplified fragments from the previous step are randomly selected according to a given value of sequencing depth, which is the total number of reads (fragments) to sequence.

(6) After the sequencing step (assuming all reads are correctly sequenced and mapped to their original gene), we can get the UMI counts for UMI protocols and read counts for non-UMI protocols.

Note that for simplicity, this pipeline omits several steps, including reverse transcription, and library cleaning up.

The runtime to simulate a dataset is dominated by the simulation of these technical steps. The total time cost varies with mRNA capture efficiency and protocol. With one CPU core of an Intel(R) Xeon(R) CPU E5–2690 v4 @ 2.60 GHz, the runtime and memory consumed to simulate a dataset with 500 cells and 10000 genes for different parameters are shown in Table 1.

**Simulation of amplification biases**. During PCR amplification of the full-length cDNAs, the PCR amplification rate (namely, probability to be amplified) can vary for different transcripts. As a result, some transcripts are over- and some are under-amplified. This causes the unwanted amplification bias. To simulate this, for each gene, its PCR amplification rate is set to a sum of a basal amplification rate (the

**Table 1 Runtime and memory usage of SymSim**

| Capture efficiency | Protocol | Sequencing depth | Runtime (min) | Memory |
|---|---|---|---|---|
| 0.05 | UMI | 100,000 | 9.5 | 1.8 G |
| 0.2 | UMI | 100,000 | 32.8 | 2.3 G |
| 0.05 | nonUMI | 100,000 | 8.6 | 1.8 G |
| 0.2 | nonUMI | 100,000 | 16.5 | 1.7 G |

**Table 2 Parameters and their ranges for simulating the UMI and nonUMI databases**

| Parameters | non-UMI database | UMI database |
| --- | --- | --- |
| Sigma (σ): within-population heterogeneity | 0.1, 0.2, 0.6 | 0.2, 0.6 |
| Gene_effects_sd: standard deviation for generating gene effect vectors | 1, 2 | 2 |
| scale_s: cell size parameter, use small values for cell types known to be small | 0.3, 0.5, 0.6, 1 | 0.1, 0.2, 0.3, 0.4, 0.5, 0.7, 1 |
| prop_hge: proportion of extreme high-expression outlier genes | 0.015, 0.02, 0.025, 0.03 | 0.01, 0.015, 0.02, 0.025, 0.03 |
| mean_hge: scale factor for high-expression outlier genes | 3, 4, 5, 6 | 3, 4, 5, 6 |
| Alpha_mean (α): mean mRNA capture efficiency | 0.05, 0.1, 0.15, 0.2, 0.25 | 0.007, 0.01, 0.02, 0.03, 0.04, 0.05, 0.07, 0.1 |
| Alpha_sd (β): standard deviation of mRNA capture efficiency across cells | 0.005, 0.01, 0.015, 0.02, 0.03, 0.045, 0.06, 0.075 | 7e−04, 0.001, 0.0014, 0.002, 0.003, 0.0035, 0.004, 0.005, 0.006, 0.007, 0.008, 0.009, 0.01, 0.012, 0.015, 0.02, 0.025, 0.03, 0.035, 0.05 |
| depth_mean (depth): mean sequencing depth for a set of cells | 1e + 05, 5e + 05, 1e + 06, 2e + 06 | 45,000, 70,000, 95,000, 150,000, 3e + 05, 5e + 05 |
| Depth_sd: standard deviation of sequencing depth for a set of cells | 100,00,30,000, 50,000, 1e + 05, 150,000, 2e + 05, 3e + 05, 6e + 05 | 4500, 7e3, 9e3, 9.5e3, 13500, 14e3, 15e3, 19e3, 21e3, 22,500, 28,500, 3e4, 35e3, 4.5e4, 47500, 5e4, 7.5e4, 9e4, 15e4, 25e4 |
| nPCR1: number of PCRs in pre-amplification phase | 14, 18 | 10, 14 |

input parameter rate_2PCR, which equals to the average amplification rate across all genes) plus a bias term B. The bias term B ranges from -MaxAmpBias to MaxAmpBias, where MaxAmpBias is a user-specified parameter to represent the total amount of amplification bias in our system.

B is composed of two categories of biases: biases related to transcript length (referred to as gene length bias, denoted by $B_{length}$) and biases caused by other factors (denoted by $B_{rand}$). We use a linear function to model the gene length bias: we first bin all gene lengths into nbins bins, and get the average length in each bin: $L = (l_{bin(1)}, l_{bin(2)}, ..., l_{bin(nbins)})$. The length bias term associated with a gene in bin i is set to: $B_{length}(i) = lenslope × median(L) − lenslope × L(i)$.

The parameter lenslope controls the extent of gene length bias. To ensure that $B_{length}$ does not exceed MaxAmpBias, the parameter lenslope should be smaller than $2 × MaxAmpBias/(nbins − 1)$. We then set the second term $B_{rand}$ to a random value in the residual range $[−MaxRandBias, MaxRandBias]$, where $MaxRandBias = MaxAmpBias − max(B_{length})$. Namely, $B_{rand} = N(0, MaxRandBias)$, where $N()$ is a Gaussian.

Therefore, for a given gene with length l, its PCR amplification rate is:

$$rate\_2PCR + B_{length}(bin(l)) + B_{rand} \qquad (1)$$

This rate is used in all rounds of PCRs in the pre-amplification step. The biases then get amplified as more PCR cycles are performed, where transcripts with higher amplification rate will likely get more molecules. Assigning a UMI to each molecule before amplification allows us to collapse all molecules with the same UMI after amplification, so different amplification rates will not affect the final molecule counts. For Fig. 4b, lenslope is set to 0.023, MaxAmpBias is set to 0.3, nbins is set to 20, and rate_2PCR is set to 0.7.

**Fitting simulation parameters to real data.** To find the best matching parameters to a real dataset, we simulate a database of datasets with a grid of parameters over a wide range. For each simulated dataset, we calculate the following statistics: mean, percent non-zero, standard deviation of genes over all cells. Then given a real dataset, we find the simulated dataset, which have the most similar distributions of the statistics to the real data, and return the corresponding parameter configurations. The parameters and their ranges for simulating the two databases are in Table 2.

**Applying dimensionality reduction and clustering methods.** We apply three different dimensionality reduction methods to cluster cells simulated from multiple discrete populations: PCA, scVI, and SIMLR. PCA is the naive baseline method that is also the most commonly seen in single-cell RNA-seq analysis. scVI is a more recent method that uses a zero-inflated negative binomial variational auto-encoder model to infer latent space for each single cell. For both the first two methods, cluster identities are then assigned using k-means clustering. The third method, SIMLR, performs dimensionality reduction and cluster identity iteratively to maximize cluster separation. The fourth method, implemented in Seurat, uses PCA for dimensionality reduction and the Louvain clustering.

**Simulation of differentially expressed genes.** Diff-EVFs give rise to differences between populations as well as DE genes between populations. DE genes by design are the ones with non-zero gene effect values corresponding to the Diff-EVFs (Fig. 6a), as the gene effect vectors are sparse with a majority of values being 0 s. Nevertheless, in some cases, the actual expression values of genes with at least one Diff-EVFs might not differ since the effects of different Diff-EVFs or the effects of modifying combinated kinetic parameters may cancel out. Differential expression might also be blurred by a high within-population variability. Thus we also use the log2 fold change (LFC) of mean gene-expression from the two populations as another criteria. The mean expression can be calculated based on simulated true counts, which is subject to gene-expression intrinsic noise, or based on the kinetic parameters themselves, directly from the theoretical gene-expression distribution. If the kinetic parameters of a gene in a cell is $k_{on}$, $k_{off}$, and s, the expected gene-expression of this gene in this cell is $s*k_{on}/(k_{on} + k_{off})$. We use multiple thresholds ranging from 0.6 to 1 on the |LFC| to define a gene is DE, in order to avoid being biased with one single artificial threshold.

**Detection of differentially expressed genes.** DE genes in observed counts are detected, respectively, with edgeR, DESeq2, Wilcoxon test, and Student t-test. For edgeR, we used the quasi-likelihood approach (QLF) with cellular detection rate (the fraction of genes that are detected with non-zero counts in each cell) as covariate. For DESeq2, we use local for the fittype parameter, and we evaluate its performance, respectively, based on the output p-values and adjusted p-values, which serve as filtering of genes.

The output from each DE method is a p-value for each gene, with smaller values meaning the gene is more likely to be a DE gene. We use two metrics to evaluate the performance of a DE method: (a) AUROC (area under receiver operating characteristic curve), where we apply different thresholds on the p-values to obtain different sets of predicted DE genes, and we can then plot ROC curves with different combinations of 1-specificity and sensitivity, thus calculate the area under the ROC curve. (b) Negative of Spearman correlation between the p-values of each detection method and the log fold difference of the true expression levels. Genes with high log fold change in true transcript counts should correspond to low p-value if the DE method works well. As the inferred p-values and log fold change in true counts are expected to be anti-correlated, we take the negative of this correlation, such that higher value corresponds to better performance.

**Applying trajectory inference methods.** We use the R packages dynwrap (https://github.com/dynverse/dynwrap, version 0.1.0) and dynmethods (https://github.com/dynverse/dynmethods, version 0.1.0) to run the three trajectory inference methods compared in this manuscript: Monocle (version 2.6.4), Slingshot (version 0.99.12), and MST (a basic method implemented in dynmethods). All methods were run with default parameters. Both dynwrap and dynmethods are under the collection of R packages dynverse used in the manuscript by Saelens et al.[52].

**Effect of parameter bimod on gene-expression levels.** To investigate if increasing bimod will cause decrease in overall gene-expression levels of genes thus lead to the decrease in performance of both clustering and DE methods, we

calculate the percentage change of total number of transcripts of genes from $bimod$ $= 0$ to $bimod = 1$. That is, for each gene, we calculate $\left( \sum_{j=1}^{m} x_j - \sum_{j=1}^{m} x'_j \right) / \sum_{j=1}^{m} x_j$, where $x_j$ is the number of transcripts of this gene in cell $j$ when $bimod = 0$, $x'_j$ is the number of transcripts of this gene in cell $j$ when $bimod = 1$, and $m$ is the number of cells. From Supplementary Fig. 10d we see that there is no consistent increase or decrease of total number of transcripts for the genes when changing $bimod$. Therefore, we conjecture that the drop in the performance of clustering and DE is rather caused by change in the distribution of gene-expression levels of genes instead of overall gene-expression levels.

**Calculating the probability of detecting a population**. Assuming all sequenced cells are correctly assigned to its original population, the probability that at least $x$ cells are detected from a population only depends on the binomial sampling. Denote the total number of cells by $N$ and the proportion of the cells in the given population by $r$, the probability that at least $x$ cells are detected for the population is:

$$1 - \sum_{k=0}^{x-1} \binom{N}{k} r^k (1-r)^{(N-k)} \qquad (2)$$

This formula is used to generate the black curves in Fig. 8a–d.

During our simulation to estimate the number of cells needed to detect a rare population, we simulate the random sampling process as follows: we start with a total of 10,000 cells for all five populations with 2000 cells for each population. We set probability vector of a cell belonging to each population to be (0.25, 0.05, 0.25, 0.25, and 0.2), where Population 2 is the rare population with smallest probability. For each randomization and given total number of cells $N$ ($N \leq 7000$), we randomly sample $N$ cells from the pool of 10,000 cells according to the probability vector.

**Reporting summary**. Further information on research design is available in the Nature Research Reporting Summary linked to this article.

## Data availability

We use four experimental datasets throughout this paper, one is from[41] which does not use UMIs and the other three use UMIs. The non-UMI datasets profiles Th17 cells under various conditions and in our paper we use a subpopulation of 130 TGF-β1 + IL-6 cells. We refer to this dataset as the "non-UMI Th17" dataset. This dataset is available at GEO with accession number GSE74833. The first UMI dataset profiles 3006 cerebral cortex cells[38]. This data are available at GEO with accession number GSE60361. The authors found nine classes in these cells. In this paper, to get distributions of kinetic parameters to map our simulated parameters to the same distribution, we perform parameter estimation, respectively, on (1) 628 cells sampled from the oligodendrocyte class; (2) 715 cells sampled from the CA1 pyramidal neurons; (3) 296 cells sampled from the S1 pyramidal neurons. To verify that SymSim can simulate data with similar statistics with given experimental dataset, we use all 948 oligodendrocyte cells in the cortex dataset. The other two UMI datasets are downloaded from the 10x Genomics website: one has around 4538 Pan T Cells (denoted as the "UMI 10x t4k" dataset, https://support.10xgenomics.com/single-cell-gene-expression/datasets/2.0.1/t_4k) and the other has 8381 PBMC cells (denoted as "UMI 10x pbmc8k", data available at https://support.10xgenomics.com/single-cell-gene-expression/datasets/2.1.0/pbmc8k). For both 10x datasets, we use cluster 1 (the largest cluster) identified at their respective analysis page. All other relevant data are available upon request.

## Code availability

SymSim is publically available as an R package on GitHub (https://github.com/YosefLab/SymSim).

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

## Acknowledgements

X.Z. was supported by grant #220558 from the Ragon Institute of MGH, MIT and Harvard. C.X. and N.Y. were supported by NIH/NHLBI grant U19 AI-090023–09.

## Author contributions

X.Z., C.X., and N.Y. conceived the study and wrote the paper. X.Z. and C.X. wrote the software and conducted the analysis.

## Additional information

**Competing interests:** The authors declare no competing interests.

