## [Peer Review File · Nature Communications]

Reviewers' comments:

Reviewer #1 (Remarks to the Author):

Summary: In this manuscript, the authors proposed a simulator (called SymSim) to generate synthetic single-cell RNA-seq (scRNA-seq) data, motivated by the kinetic model of stochastic gene expression. They incorporated three main sources of variability with the simulator: extrinsic biological noise driven by heterogeneous cell states, intrinsic biological noise generated by the stochastic nature of gene expression in a single cell, and technical noise generated by inefficient mRNA capture and amplification. Using two scRNA-seq datasets with or without UMIs, they showed that the statistical properties of the real data are well matched to the simulated data. The simulator was applied to evaluate the widely used statistical methods for clustering and detecting differentially expressed genes, and to choose the optimal number of cells.

The present manuscript makes a valuable contribution by proposing a unified framework for simulating synthetic scRNA-seq data, which will be useful for evaluating computational methods for analyzing scRNA-seq data. However, it was tested on very limited datasets and its applicability is not properly demonstrated in this manuscript.

Major points:

1. The authors used three statistics as measures for comparison between simulated and real data: mean, percent-non-zero, and standard deviation. However, these three measures were already used to fit simulation parameters to real data, which suggests the possibility of unfair comparison. The comparison should be done based on independent measures not used in the process of fitting (e.g. mean-variance relationship (<https://www.ncbi.nlm.nih.gov/pubmed/28899397>), goodness of fit statistic (<https://www.ncbi.nlm.nih.gov/pubmed/29036287>)).
2. In general, the simulators play an important role in evaluating the performance of computational methods for analyzing scRNA-seq data. The previous simulators such as Splatter and powsimR focused on the problems of clustering, detecting differentially expressed genes, and power analysis. It is timely to consider a unified framework for generating synthetic scRNA-seq data for evaluating the performance of computational tools for reconstructing differentiation trajectories, correcting for batch effects, and imputing sparse scRNA-seq data for inferring gene regulatory networks, which will have a high impact and well appreciated in the field of single-cell genomics. The idea of incorporating extrinsic biological variability through extrinsic variability factors suggests that SymSim can be directly used to provide synthetic scRNA-seq data for trajectory and imputation analysis. I strongly suggest that the authors should apply their simulator to these problems (at least trajectory analysis), which will greatly improve the impact of this manuscript.
3. The authors showed that their simulator outperforms Splatter from two datasets. They should also include powsimR (<https://www.ncbi.nlm.nih.gov/pubmed/29036287>) since a comparison study demonstrated that powsimR outperform Splatter in a subset of performance measures (<https://www.ncbi.nlm.nih.gov/pubmed/29028961>). Additionally, more real data covering diverse scRNA-seq protocols should be used for comparison to reach a general conclusion (e.g. powsimR used 38 datasets).
4. Estimating kinetic parameters from real data: The simulator first estimates kinetic parameters from real data based on the formulation of Kim et al. [26], which then be used to simulate extrinsic variability. The underlying generative model at this step does not properly capture the technical variability of scRNA-seq data since the generative model only assumes the Binomial sampling with the fixed "f" (which corresponds to the Poisson noise). This means that the intrinsic biological variability

generated by stochastic gene expression is over-estimated since the technical noise modeled by the Poisson noise is severely underestimated. It might be possible that the simulated extrinsic and intrinsic variability are not properly captured and the interpretability of parameters of SymSim is limited.

5. It is not clear to me why EVFs (non-negative) and capture efficiency alpha (its support is between 0 and 1) are sampled from normal distributions instead of gamma and beta distribution?

6. It might be useful to draw a full generative process of SymSim using a graphical representation specifying random variables and their distributions.

Reviewer #2 (Remarks to the Author):

Summary

In this paper the authors develop a method to simulate scRNA-seq count matrices using three sources of variation: transcriptional variation, a low rank factor model of biological variation (further constrained by a tree structure), and noise representing the measurement process.

This method does seem like an improvement over Splatter or the more specialized PowsimR, and can be of use to the field to evaluate computational methods.

There are still some concerns however, in particular regarding similarity with real data.

Major issues

- Simulations of UMI counts are only compared to STRT-seq data, rather than the by far more commonly used droplet based 10X Genomics Chromium method. Including other data types is fine, but all evaluations must also be in terms of 10X data for the simulation to be of use to the community at large.
- Parameters are fitted to imputed data, which is not a valid practice. Any variation in the data will just reflect the assumptions of the imputation model, and not true variation.
- Several plots describing summary statistics for real data look wrong. For example, $\log(\text{mean})$ only goes down to 0, but in real data the log of the mean can be as small as $-\log(N_{\text{cells}})$. In particular the figure S3c indicates problems with the real data, there seem to be no biological variation at all in the real data. Was the comparison to real data done using imputed data?
- Distributions of many more summary statistics for data should be compared between simulated and real data, and also compared with Splatter (when possible). With at least: number of detected genes per cells; reads per cell; reads per gene; UMIs per cell; UMIs per gene; coefficient of variation per gene; fragment length distribution; PCR duplication per gene (num reads / num umis); num umis per cell sorted by rank (the typical 10X QC plot for identifying real cells).
- What aspects of this simulator contributes most to the increased fit compared to Splatter? The technical, transcriptional, or biological components?

Minor issues

- It is clear from the text that the clustering comparison is a proof of concept rather than an exhaustive comparison. But K-means is very rarely used in the field, it would be good to at least include the very popular graph based clustering in Seurat.
- The sigma parameter is found to have the largest effect on clustering performance. This parameter however seem to be the one that is the hardest to find an interpretation for in real data, and does not seem possible to learn or estimate from real data, and even more so in the phase of designing an experiment. Discussion about this parameter is warranted.
- All variables that can be set in the simulation and mentioned in the text should have units described. For example, the k_{on} parameter would have unit mRNA's / second?
- There seem to be an issue with the JAGS inference of the parameters. MCMC should not depend on initial condition, contrary to what methods section mention. If MCMC chains have autocorrelation it is an indicator that the model did not fit correctly and the results should not be used.
- Simulating capture efficiency from a normal as is described does not make sense since it can never be negative. In the R implementation small values from the draws are assigned the value 0.0005, making this single value much more likely to be drawn than any other value. (A proper truncation would be to discard and resample if the drawn value is smaller than 0.0005). There is also no motivation for this cutoff, scRNA-seq is known to be very inefficient.
- For the DE comparison, it should be written out what "sufficiently large fold change" would be.
- The metrics for comparing DE results are not clear, they need more description and motivation.
- While the DE comparison is a proof of concept and not exhaustive, it would be interesting to see performance on doing DE with uneven population, e.g. 30 vs 300 cells. This is more in line with the common task of finding markers for cell types.
- The sample size evaluation does show that the minimal number of cells from the Satija website is not enough, the SymSim analysis doesn't consider the random sampling of cells from a larger population at all, making the comparison hard. They are measuring different things, ideally both should be considered in one step.
- In the edgeR "cellular detection rate" is used as a covariate, but it is not described what this means.

Reviewer #3 (Remarks to the Author):

Here the authors present a program to simulate single cell RNA-seq data based on sets of input parameters with intuitive biological or technical interpretations, such as mRNA-capture efficiency or the probabilities of switching a promoter on or off.

The interpretability of these parameters is advertised as one of the main advantages of SymSim as compared to others that are based on pure empirical distributions. However, the crux with those parameters is that they are by in large unknown. Therefore, the authors also have a method to estimate parameters of the kinetic part of the model and probably also the same is done for the technical variables (I found one mentioning of capture efficiency in the Discussion). It is indeed

interesting to quantify the parameters of a bursting model of transcription for different types of cells and genes.

My main criticism of this paper is that the authors fail to show that they manage to deconvolve the parameters of the kinetic model from the technical variation. They only state that their estimates are consistent with previous estimates, which is not so surprising, since the data and the model are basically the same. Since the authors already generated a large database of parameter-combinations a proper analysis of the validity of the parameter estimates should be straightforward. They just need to check how well they can recover the parameters of the kinetic model, given various levels of technical noise. If the parameters can be recovered this is proof that the parameters are indeed interpretable according to the model, otherwise they are not more helpful than an empirical distribution.

Furthermore, the importance of having a kinetic model is motivated, by claiming that the potential bimodality of the resulting expression values might have an impact on clustering and differential expression analysis. However this argument is not picked up in the analysis. Theoretically their Simulations should allow them to assess the impact of k_{on}/k_{off} on the power to detect cell-types and DE-genes.

Finally, it is common to compare a new method to already existing ones and the authors here choose to compare SymSim to splatter. SymSim produces more realistic datasets than splatter, however Sonesson et al 2018(<http://dx.doi.org/10.1093/bioinformatics/btx631>) show splatter is not the best scRNA-seq available: powsimR (Vieth et al. 2017) simulates much more realistic scRNA-seq than splatter, hence SymSim should be also compared to powsimR.

Minor comments:

Brownian motion on a tree is not per se new, the authors should cite previous work. For example, there is a whole chapter on this in the Felsenstein book *Inferring Phylogenies* and I believe that it also has been applied in the analysis of RNA-seq data.

If also the elements of technical variation are estimated, this should be clarified in the text.

The comparison of the 4 DE-methods is a bit half-hearted (only 4 methods, of which 2 are only applicable to pairwise comparisons + a fairly ad hoc LFC cutoff). It was briefly mentioned that Sonesson et al. 2018 gave similar conclusions. All in all I don't see the added value of the presented DE analysis.

What are the run-times and memory requirements for the simulations and the fit the dataset to the database are missing.

P.20 I have never seen 'self-explanatory' input parameters.

At some point the authors start to make general statements about UMI and non-UMI data. If I understood correctly the authors have only analysed 1 UMI and 1 non-UMI data-set and hence this generalization is not justified.

P.20 References for trajectory & RNA-velocity analysis are missing

Since the authors advertise SymSim to be good for scRNA-seq method evaluation, they should also discuss Ziegenhain et al (2017) who do exactly this using an early version of powsimR.

Response to reviewers' comments are in blue

Changes in manuscript are in red

Response to Reviewers' Comments

Reviewer #1 (Remarks to the Author):

Summary: In this manuscript, the authors proposed a simulator (called SymSim) to generate synthetic single-cell RNA-seq (scRNA-seq) data, motivated by the kinetic model of stochastic gene expression. They incorporated three main sources of variability with the simulator: extrinsic biological noise driven by heterogeneous cell states, intrinsic biological noise generated by the stochastic nature of gene expression in a single cell, and technical noise generated by inefficient mRNA capture and amplification. Using two scRNA-seq datasets with or without UMIs, they showed that the statistical properties of the real data are well matched to the simulated data. The simulator was applied to evaluate the widely used statistical methods for clustering and detecting differentially expressed genes, and to choose the optimal number of cells.

The present manuscript makes a valuable contribution by proposing a unified framework for simulating synthetic scRNA-seq data, which will be useful for evaluating computational methods for analyzing scRNA-seq data. However, it was tested on very limited datasets and its applicability is not properly demonstrated in this manuscript.

Major points:

1. The authors used three statistics as measures for comparison between simulated and real data: mean, percent-non-zero, and standard deviation. However, these three measures were already used to fit simulation parameters to real data, which suggests the possibility of unfair comparison. The comparison should be done based on independent measures not used in the process of fitting (e.g. mean-variance relationship (<https://www.ncbi.nlm.nih.gov/pubmed/28899397>), goodness of fit statistic (<https://www.ncbi.nlm.nih.gov/pubmed/29036287>)).

Following this comment and a comment by Reviewer 2, we have compared the simulated and real datasets using additional measures, including: coefficient of variation of genes (Figure S5d), mean-variance relationship (Figure S6a), mean vs percent-non-zero (percentage of cells expressing this gene, Figure S6b), distribution of number of detected genes per cell (Figure S7a), distribution of number of UMIs per cell (Figure S7b), and number of UMIs per cell sorted by rank (Figure S7c). We have also included another published simulator powSimR (Vieth et al. 2017) in the comparisons wherever applicable. SymSim shows the best overall performance in terms of all additional measures suggested by Reviewers 1 and 2 compared to published simulators Splatter and powSimR. We believe that these results strengthen our conclusion that SymSim provides simulated datasets which resemble the given real datasets.

Notably, The goodness-of-fit measure suggested above, which is usually used to compare a set of samples against a given probability distribution, is not applicable for our case, as we are comparing simulated data against real data.

2. In general, the simulators play an important role in evaluating the performance of computational methods for analyzing scRNA-seq data. The previous simulators such as Splatter and powsimR focused on the problems of clustering, detecting differentially expressed genes, and power analysis. It is timely to consider a unified framework for generating synthetic scRNA-seq data for evaluating the performance of computational tools for reconstructing differentiation trajectories, correcting for batch effects, and imputing sparse scRNA-seq data for inferring gene regulatory networks, which will have a high impact and well appreciated in the field of single-cell genomics. The idea of incorporating extrinsic biological variability through extrinsic variability factors suggests that SymSim can be directly used to provide synthetic scRNA-seq data for trajectory and imputation analysis. I strongly suggest that the authors should apply their simulator to these problems (at least trajectory analysis), which will greatly improve the impact of this manuscript.

This is a very good point. Indeed, the ability of generating given structure of continuous populations of SymSim makes it a natural choice to benchmark trajectory inference methods. In this revision, we added Figure 7, where we show the comparison of three trajectory inference methods: Monocle DDRTree, Slingshot, and Minimum Spanning Tree. Please refer to the section “Using SymSim to evaluate methods for clustering, differential expression and trajectory inference” and Figure 7 legend for details of our analysis.

3. The authors showed that their simulator outperforms Splatter from two datasets. They should also include powsimR (<https://www.ncbi.nlm.nih.gov/pubmed/29036287>) since a comparison study demonstrated that powsimR outperform Splatter in a subset of performance measures (<https://www.ncbi.nlm.nih.gov/pubmed/29028961>). Additionally, more real data covering diverse scRNA-seq protocols should be used for comparison to reach a general conclusion (e.g. powsimR used 38 datasets).

We thank the reviewer for pointing us to the references. For a more comprehensive comparison with published simulators, we added comparison with powsimR, when we compare the distribution of mean, percent-non-zero, standard deviation and coefficient of variation between real data and simulated data (Figures S5c-d). We also included powsimR in the comparison of mean-variance relationship (Figure S6a), mean-percent-nonzero relationship (Figure S6b),

distribution of the number of expressed genes per cell (Figure S7a), distribution of the number of UMIs per cell (Figure S7b) and number of UMIs per cell sorted by rank (Figure S7c).

To test SymSim on additional cases, we took two datasets from 10x Genomics - a popular platform for large scale single cell transcriptome sequencing. One 10x dataset consists of 4538 human T Cells (denoted as the “t4k” dataset, data available at https://support.10xgenomics.com/single-cell-gene-expression/datasets/2.0.1/t_4k) and the other consists of 8381 human PBMC cells (denoted as pbmc8k, data available at <https://support.10xgenomics.com/single-cell-gene-expression/datasets/2.1.0/pbmc8k>). We have included the results on the t4k dataset in Figure 4e and that on the pbmc8k dataset in Figure S5b. As the representative dataset from the 10x platform, dataset t4k is included in all follow-up comparisons in Figures S5c-d, S6a-b and S7a-c. Overall, SymSim provides simulated datasets which resemble both 10x datasets well and better than the benchmark simulators.

Notably, we noticed that the 10x datasets include a small proportion of outlier genes that are present in a large proportion of the cells and with an unusually high level of expression. This phenomenon is more pronounced in the 10x data (e.g., compared to the Smart-Seq2), possibly due to a lower sensitivity. To address this, Splatter (Zappia, Phipson, and Oshlack 2017) introduced additional parameters to adjust the expression levels specifically for these genes. To gain more flexibility in this respect, we also added two extra parameters to SymSim to model these outlier genes. First, we introduced a parameter *prop_hge*, which represents the proportion of outlier (highly expressed) genes. We then assume that these genes are constitutively transcribed and thus that their expression values are generated by a Poisson distribution with a parameter s (the transcription rate). The parameter s for each outlier gene is calculated from the EVF vectors and gene effect vectors, as in the other genes. To reflect their high expression levels, we further multiply each rate s by an inflation factor, generated from a linear function defined by a second parameter *mean_hge* (>1). The revised manuscript includes a description of this addition (Section “The first knob: allele intrinsic variation” and Methods).

To maintain consistency with this minor modification to our model, we re-generated all related figures (Figures 2-8 in the main text, and Figures S2-S4, S8-S11 in the supplementary figures). All reproduced figures show high consistency with the original ones.

4. Estimating kinetic parameters from real data: The simulator first estimates kinetic parameters from real data based on the formulation of Kim et al. [26], which then be used to simulate extrinsic variability. The underlying generative model at this step does not properly capture the technical variability of scRNA-seq data since the generative model only assumes the Binomial sampling with the fixed “ f ” (which corresponds to the Poisson noise). This means that the intrinsic biological variability generated by stochastic gene expression is over-estimated since the technical noise modeled by the Poisson noise is severely underestimated. It might be possible that the simulated extrinsic and intrinsic variability are not properly captured and the interpretability of parameters of SymSim is limited.

The reviewer’s major concern here is that we do not get accurate estimation of kinetic parameters from real data. We would like to point out the following:

- 1) As we discuss in the manuscript, we do not claim to infer the accurate parameters for every cell and gene since (in the absence of additional assumptions) it is a very challenging problem, which can be ill defined. Instead, we aim to infer a reasonable distribution for the values of each kinetic parameter. We then use these distributions to re-scale the kinetic parameters calculated from the EVF vectors and gene effect vectors to fit into realistic value ranges (Figure 2a).
- 2) To decrease the effect of biological variability in real data, we conduct estimation of these distributions only considering relatively homogenous subsets of cells (i.e., clusters). To further decrease the effects of technical noise within these sub-populations we impute the zero entries using scVI and MAGIC (which can often be ascribed to low sensitivity [see (Lopez et al. 2018; van Dijk et al. 2018)]). Finally, to increase robustness, we repeated the inference procedures with several such sub-populations and use an aggregate over all the inferred distributions.
- 3) To further increase our confidence that the inferred parameter ranges are realistic, we conducted literature curation, and documented experimentally- measured kinetic coefficients (Supplementary Materials, Section 1), showing that they are within our inferred range.
- 4) As further support, we show that using a single estimation for the distribution of each kinetic parameter (Figure 2b), we can generate expression profiles that fit different data sets quite accurately and better than existing simulators (Figures 4e and S5).

- 5) Finally, as suggested by Reviewer 3, we performed simulation to test if our procedure of kinetic parameter estimation can reconstruct the ranges of true parameters, and in particular, how does the estimation obtained using imputed counts compared to that using true counts (which does not have technical variability). Figure S3a shows that imputed counts and true counts give comparable estimation, and both result in estimations with similar ranges to true parameters.

We have updated the text in the manuscript in Section “The first knob: allele intrinsic variation” (Pages 5-6) and Figure 2a in order to avoid any confusion.

5. It is not clear to me why EVFs (non-negative) and capture efficiency α (its support is between 0 and 1) are sampled from normal distributions instead of gamma and beta distribution?

The EVFs are latent variables corresponding to biological conditions which determine the identity of each cell. The values of the variables can be rather abstract, and we do not have specific reasons to believe that they follow a certain distribution. Neither do we enforce them to be non-negative. In this case, Normal distribution is simply a natural choice.

In the case of capture efficiency α , we understand that the benefit of using Beta distribution is that the values fall into the range of [0,1]. However, when we have the mean α value (which is usually less than 0.2), we want the α values for all cells to center around the mean with a certain (small) standard deviation instead of having all values spanning through the region between 0 and 1. Of course, we can scale the Gamma or Beta distributions for this purpose. For now we keep our implementation with truncated Gaussian because we do not see significant advantage of using other distributions. In our future release of the package we are happy to add different choices of distributions for users to choose from.

6. It might be useful to draw a full generative process of SymSim using a graphical representation specifying random variables and their distributions.

We thank the reviewer for this advice. We have drawn a generalized version of the right part of Figure 1, which includes the workflow of SymSim and the parameters involved at each step as well as the corresponding distributions for parameters. This more detailed illustration is in Figure S1.

Reviewer #2 (Remarks to the Author):

Summary

In this paper the authors develop a method to simulate scRNA-seq count matrices using three sources of variation: transcriptional variation, a low rank factor model of biological variation (further constrained by a tree structure), and noise representing the measurement process.

This method does seem like an improvement over Splatter or the more specialized PowsimR, and can be of use to the field to evaluate computational methods.

There are still some concerns however, in particular regarding similarity with real data.

Major issues

1. Simulations of UMI counts are only compared to STRT-seq data, rather than the by far more commonly used droplet based 10X Genomics Chromium method. Including other data types is fine, but all evaluations must also be in terms of 10X data for the simulation to be of use to the community at large.

We thank the reviewer for the excellent suggestion. In the revised version, we have included two 10x datasets to evaluate SymSim. One 10x dataset consists of 4538 human T Cells (denoted as the “t4k” dataset, data available at https://support.10xgenomics.com/single-cell-gene-expression/datasets/2.0.1/t_4k) and the other consists of 8381 human PBMC cells (denoted as pbmc8k, data available at <https://support.10xgenomics.com/single-cell-gene-expression/datasets/2.1.0/pbmc8k>). We compare SymSim (using these datasets in addition the ones used in the previous submission) to two published simulators: Splatter (which we used previously) and powsimR (added in this revision) in terms of their consistency with the real data. Specifically, we have included the results on the t4k dataset in Figure 4e and that on the pbmc8k dataset in Figure S5b. As the representative dataset from the 10x platform, dataset t4k is included in all follow-up

comparisons in Figures S5c-d, S6a-b and S7a-c. Overall, SymSim provides simulated datasets which resemble both 10x datasets well and better than the benchmark simulators.

Notably, in order to account for outlier genes with high UMI counts in the 10x datasets, we added two parameters to SymSim. For details, please refer to our response to point 3 of Reviewer 1's comments (Pages 3-4 of this document), and Sections "The first knob: allele intrinsic variation" and Methods in the manuscript.

2. Parameters are fitted to imputed data, which is not a valid practice. Any variation in the data will just reflect the assumptions of the imputation model, and not true variation.

First, it is important to note that the imputed data is only used to infer the distributions of each kinetic parameter, which are used to scale the dot-products into realistic ranges (Pages 5-6 in the manuscript). These inferred distributions are fixed (Figure 2b), and not being re-learned for every new dataset that we analyzed. When we fit the SymSim model to new data we therefore use the observed counts without applying any additional imputation.

Regarding the validity of the inference of plausible distributions of each kinetic parameters:

1) As we discuss in the manuscript, we do not claim to infer the accurate parameters for every cell and gene since (in the absence of additional assumptions) it is a very challenging problem, which can be ill defined. Instead, we aim to infer a reasonable distribution for the values of each kinetic parameter. We then use these distributions to re-scale the kinetic parameters calculated from the EVF vectors and gene effect vectors to fit into realistic value ranges (Figure 2a).

2) To decrease the effect of biological variability in real data, we conduct estimation of these distributions only considering relatively homogenous subsets of cells (i.e., clusters). To further decrease the effects of technical noise within these sub-populations we impute the zero entries using scVI and MAGIC (which can often be ascribed to low sensitivity [see (Lopez et al. 2018; van Dijk et al. 2018)]). Finally, to increase robustness, we repeated the inference procedures with several such sub-populations and use an aggregate over all the inferred distributions.

3) To further increase our confidence that the inferred parameter ranges are realistic, we conducted literature curation, and documented experimentally- measured kinetic coefficients (Supplementary Materials, Section 1), showing that they are within our inferred range.

4) As further support, we show that using a single estimation for the distribution of each kinetic parameter (Figure 2b), we can generate expression profiles that fit different data sets quite accurately and better than existing simulators (Figures 4e and S5).

5) Finally, as suggested by Reviewer 3, we performed simulation to test if our procedure of kinetic parameter estimation can reconstruct the ranges of true parameters, and in particular, how does the estimation obtained using imputed counts compared to that using true counts (which does not have technical variability). Figure S3a shows that imputed counts and true counts give comparable estimation, and both result in estimations with similar ranges to true parameters.

We have updated the text in the manuscript in Section “The first knob: allele intrinsic variation” (Pages 5-6) and Figure 2a in order to avoid any confusion.

3. Several plots describing summary statistics for real data look wrong. For example, $\log(\text{mean})$ only goes down to 0, but in real data the log of the mean can be as small as $-\log(N_{\text{cells}})$. In particular the figure S3c indicates problems with the real data, there seem to be no biological variation at all in the real data. Was the comparison to real data done using imputed data?

When we plot $\log(\text{mean})$, it is on the gene-expression after adding one. We apologize for not being accurate with the figure legends. We have updated the corresponding legends to $\log(\text{mean}+1)$.

Figure S3c is now included in Figure S6b in the new Supplementary figures file. Figure S6b shows the pattern of percentage of cells expressing a gene *versus* the mean expression of the gene. The previous Figure S3c only plots the red dots in the new Figure S6b, which are the percentiles of genes instead of the genes themselves. The blue dots in Figure S6b correspond to genes, and we believe this would make more sense to the readers.

Throughout the manuscript, imputed data is used only to estimate the distributions of kinetic parameters shown in Figure 2b, which are used to adjust the kinetic parameters calculated from EVFs and gene effect vectors into realist ranges.

4. Distributions of many more summary statistics for data should be compared between simulated and real data, and also compared with Splatter (when possible). With at least: number of detected genes per cells; reads per cell; reads per gene; UMIs per cell; UMIs per gene; coefficient of variation per gene; fragment length distribution; PCR duplication per gene (num reads / num umis); num umis per cell sorted by rank (the typical 10X QC plot for identifying real cells).

We thank the reviewer for suggesting a very comprehensive set of measurements to evaluate the simulators. We have added the results on comparison between the three simulators (SymSim, Splatter, powSimR) with the following measurements: coefficient of variation of genes (Figure S5d), mean vs percent-non-zero (percentage of cells expressing this gene, Figure S6b), mean-variance relationship (suggested by Reviewer 1, Figure S6a), distribution of number of detected genes per cell (Figure S7a), distribution of number of UMIs per cell (Figure S7b), number of UMIs per cell sorted by rank (Figure S7c). SymSim shows the best overall performance in terms of all these additional measures compared to published simulators Splatter and powSimR.

Reads per gene and UMIs per gene were already shown in our submitted version as mean gene-expression in Figure 4e and Figures S5b-c. Splatter and powsimR do not output information to plot “fragment length distribution” and “PCR duplication per gene (num reads / num umis)” so we do not compare these measures between the simulators.

5. What aspects of this simulator contributes most to the increased fit compared to Splatter? The technical, transcriptional, or biological components?

As the reviewer mentioned, the added accuracy can arise from different components of the model, from the fitting procedure, or from various combinations thereof. The underlying factors for increased performance may also vary between datasets. In practice, it can therefore be difficult to ascribe improvement in performance to specific components. Instead, in our discussions we describe the conceptual differences between SymSim and other simulators, and the pertaining effects on usability (e.g., being able to test the effects of technical parameters of library construction) and accuracy (in terms of fitting to real data).

Minor issues

- It is clear from the text that the clustering comparison is a proof of concept rather than an exhaustive comparison. But K-means is very rarely used in the field, it would be good to at least include the very popular graph based clustering in Seurat.

We thank the reviewer for this suggestion. In our revised version, we have included Seurat when we comparing the clustering methods under different parameter settings in Figures 5b-c. Seurat is shown to be better than SIMLR and PCA+kmeans, and comparable with or inferior to scVI+kmeans. We also included Seurat in Figure 8, for analysis of how many cells are needed to detect a rare population. As in this analysis we particularly require low false positives (high precision), Seurat performs better than other methods in many cases, as kmeans suffers from high false positives for the rare population.

- The sigma parameter is found to have the largest effect on clustering performance. This parameter however seem to be the one that is the hardest to find an interpretation for in real data, and does not seem possible to learn or estimate from real data, and even more so in the phase of designing an experiment. Discussion about this parameter is warranted.

The Sigma parameter corresponds to the extent of heterogeneity within a population. It is therefore the case that Sigma has a direct interpretation, but as the reviewer mentioned - it is difficult to estimate from real data. Instead, we advocate for exploration of wide range of Sigma values, which (together with the tree “weights”) give rise to a range of data characteristics - from well separated populations, to almost entirely mixed ones. In all of our benchmarks, we considered a range of these values, under the premise that they cover the relevant (physiological) range. We now clarify this point in Section “Using SymSim to evaluate computational methods for single cell RNA-seq data” (Page 16).

- All variables that can be set in the simulation and mentioned in the text should have units described. For example, the k_{on} parameter would have unit mRNA's / second?

We fix degradation rate d to constant value of 1 and consider the other three parameters (k_{on} , k_{off} and s) relative to d . This practice is used in existing work (Peccoud and Ycart 1995), (Larson 2011). We clarify this point in the first paragraph of Section “The first knob: allele intrinsic variation”.

- There seem to be an issue with the JAGS inference of the parameters. MCMC should not depend on initial condition, contrary to what methods section mention. If MCMC chains have

autocorrelation it is an indicator that the model did not fit correctly and the results should not be used.

Following this comment, we have added Figures S2c-d and modified the text in Section “Estimating kinetic parameters from real data” in Methods for better clarity.

Theoretically, MCMC should not depend on initial conditions. In practice however, bad starting values can lead to slow convergence and worse autocorrelation. We used multiple starting points for diagnose purpose. For example, our results from three different runs for the same data (UMI cortex data, subpopulation pyramidal CA1, imputed with scVI) have very similar distributions (Figure S2c). We also investigate the autocorrelation and plotted the typical lag vs correlation plots for these chains (Figure S2d). The correlation drops as the lag increases, and eventually becomes very small. We then merge all the chains to get the final distributions.

- Simulating capture efficiency from a normal as is described does not make sense since it can never be negative. In the R implementation small values from the draws are assigned the value 0.0005, making this single value much more likely to be drawn than any other value. (A proper truncation would be to discard and resample if the drawn value is smaller than 0.0005). There is also no motivation for this cutoff, scRNA-seq is known to be very inefficient.

We have updated the truncation method as suggested by the reviewer: when we get a value smaller than the threshold we resample until the value is above the threshold. Notably, we chose the value of $5e-4$ as a sensible lower bound, which fits much of the observed data. For instance, for a representative cell with 500k transcripts, this rate means 250 UMIs, which is considered very low on the sensitivity spectrum in current datasets. We therefore believe that this value is small enough and the effect it has on the final output dataset is very minor.

- For the DE comparison, it should be written out what “sufficiently large fold change” would be.

We have used multiple thresholds for the fold change ranging from 0.6 to 1 and the final performance is the average over different thresholds. We have added in parenthesis “(threshold of absolute \log_2 fold change ranges from 0.6 to 1, details in Figure legends; Methods)” after this phrase in the manuscript, and in Figure legend of Figure 6 we list the threshold used for each plot.

- The metrics for comparing DE results are not clear, they need more description and motivation.

We have added detailed description of the two metrics “AUROC” and “negative of Spearman correlation” in Methods, Section “Detection of differentially expressed (DE) genes”.

- While the DE comparison is a proof of concept and not exhaustive, it would be interesting to see performance on doing DE with uneven population, e.g. 30 vs 300 cells. This is more in line with the common task of finding markers for cell types.

This is a good point. We have included the results for uneven population sizes (30 vs 300 cells) in Figure 6d-e and Figure S9c. The performance in this case appears to be in between of the “30 vs 30” and “300 vs 300” cases.

- The sample size evaluation does show that the minimal number of cells from the Satija website is not enough, the SymSim analysis doesn’t consider the random sampling of cells from a larger population at all, making the comparison hard. They are measuring different things, ideally both should be considered in one step.

We do consider and simulate the random sampling of cells. We realize that this is not clearly described in the manuscript, so we have added description of this procedure in Methods, Section “Binomial model to calculate the probability of detecting a population”. We start with a total of 10000 cells for all five populations with 2000 cells for each population. We set probability of a cell belonging to each population as (0.25, 0.05, 0.25, 0.25, 0.2), where Population 2 is the rare population with smallest probability. For each randomization and given total number of cells N ($N \leq 7000$), we randomly sample cells from the pool of 10000 cells according to the probability vector (0.25, 0.05, 0.25, 0.25, 0.2).

- In the edgeR “cellular detection rate” is used as a covariate, but it is not described what this means.

We added the description of “cellular detection rate” as “the fraction of genes that are detected with nonzero counts in each cell” in Section “Detection of differentially expressed (DE) genes” in Methods.

Reviewer #3 (Remarks to the Author):

Here the authors present a program to simulate single cell RNA-seq data based on sets of input parameters with intuitive biological or technical interpretations, such as mRNA-capture efficiency or the probabilities of switching a promoter on or off.

The interpretability of these parameters is advertised as one of the main advantages of SymSim as compared to others that are based on pure empirical distributions. However, the crux with those parameters is that they are by in large unknown. Therefore, the authors also have a method to estimate parameters of the kinetic part of the model and probably also the same is done for the technical variables (I found one mentioning of capture efficiency in the Discussion). It is indeed interesting to quantify the parameters of a bursting model of transcription for different types of cells and genes.

In SymSim, we consider that the gene-expression level of a gene in a cell is determined by the cell identity (represented by the EVFs) and the gene identity (represented by the gene effect vectors). The kinetic parameters used in the simulation are calculated from the EVFs and gene effect vectors, and then re-scaled with a fixed set of kinetic parameter distributions shown in Figure 2b, which are inferred from experimental data (Figure 2a, Paragraphs 5-6 of Section “The first knob: allele intrinsic variation”, Section “Estimating kinetic parameters from real data” in Methods). Kinetic parameter estimation is only performed to obtain the distributions in Figure 2b.

Therefore, when we fit SymSim to a given experimental dataset, instead of fitting all kinetic parameters (which are a large number of parameters) we fit a small number of parameters which are used to generate the identify of cells and genes. In Methods (Section “Fitting simulation parameters to real data”) of the manuscript we include a whole list of parameters which are fit for every given experimental dataset in order to generate similar simulated data, as well as the interpretation of each parameter. Furthermore, we have added Figure S1 where parameters used in each step are shown.

My main criticism of this paper is that the authors fail to show that they manage to deconvolve the parameters of the kinetic model from the technical variation. They only state that there

estimates are consistent with previous estimates, which is not so surprising, since the data and the model are basically the same. Since the authors already generated a large database of parameter-combinations a proper analysis of the validity of the parameter estimates should be straight forward. They just need to check how well they can recover the parameters of the kinetic model, given various levels of technical noise. If the parameters can be recovered this is proof that the parameters are indeed interpretable according to the model, otherwise they are not more helpful than an empirical distribution.

Kinetic parameter estimation is not performed for every given experimental dataset. It is performed prior to any simulation to obtain distributions of kinetic parameters with realistic ranges, to re-scale the kinetic parameters calculated from the EVF vectors and gene effect vectors (Figure 2a).

As for the Kinetic parameter estimation step itself, as we discuss in the manuscript, we do not claim to infer the accurate parameters for every cell and gene since (in the absence of additional assumptions) it is a very challenging problem, which can be ill defined. Instead, we aim to infer a reasonable distribution for the values of each kinetic parameter. For robustness, these distributions are inferred as a consensus of multiple forms of analysis (datasets, imputation algorithms).

Following the reviewer's suggestion, we have done the following:

1. Added simulation analysis where we use the Beta Poisson model to infer the range of kinetic parameters in data from a homogenous population of cells generated by SymSim (i.e., where the ground truth is known). We use both the true transcript counts (with no technical variability added) and observed (using the same technical variability parameters as in Figure 4e [cortex dataset]). Figure S3a shows that imputed counts and true counts give comparable estimation, and both result in estimations with similar ranges to true parameters.
2. Revised the text in Section "The first knob: allele intrinsic variation" to further clarify these points.

We would also like to point out the following regarding the kinetic parameter estimation process:

- 1) To decrease the effect of biological variability, we estimate these distributions only considering relatively homogenous subsets of cells (i.e., clusters). To further decrease

the effects of technical noise within these sub-populations we impute the zero entries (which can often be ascribed to low sensitivity (Lopez et al. 2018; van Dijk et al. 2018)) using both scVI and MAGIC. Finally, to increase robustness, we repeated the inference procedures with several such sub-populations and use an aggregate over all the inferred distributions.

- 2) As further support, we show that using a single estimation for the distribution of each kinetic parameter (Figure 2b), we can generate expression profiles that fit different data sets quite accurately and better than existing simulators (Figures 4e and S5-7).
- 3) To further increase our confidence that the inferred parameter ranges are realistic, we conducted literature curation, and documented experimentally- measured kinetic coefficients (Supplementary Materials, Section 1), showing that they are in line with our inferred range.

Furthermore, the importance of having a kinetic model is motivated, by claiming that the potential bimodality of the resulting expression values might have an impact on clustering and differential expression analysis. However this argument is not picked up in the analysis. Theoretically their Simulations should allow them to assess the impact of k_{on}/k_{off} on the power to detect cell-types and DE-genes.

This is a good point. To investigate the effect of bimodality on the performance of clustering and differential expression algorithms, we increased parameter *bimod* from 0 to 1 for the dataset used in our benchmark analysis of these methods in Figures 5b-c and 6d-e, and performed clustering and DE again. The comparison of clustering results between different methods and different values of *bimod* is shown in Figure S8a. We see that most of the time there is a decrease of performance for the same method with increase of *bimod*. We then aggregate all the values of adjusted rand index for all methods and all the parameters of α and σ , but only group by the *bimod* value, and performed Wilcoxon test between the two groups of values. We can see that the difference between these two groups is small but significant ($p\text{-value} < e^{-12}$, Figure S8b).

We show the comparison of results from the differential expression methods in Figure S10. From these figures, we also see a drop in performance when increasing *bimod*, especially when

the number of cells are small. The drop is less prominent when the number of cells are 300 vs 300 compared to the other two cases in Figures S10a-b.

To investigate if increasing *bimod* will cause decrease in overall gene-expression levels of genes thus lead to the decrease in performance, we calculate the percentage change of total number of transcripts of genes from *bimod*=0 to *bimod*=1. That is, for each gene, we calculate $(\sum_{j=1}^m x_j - \sum_{j=1}^m x'_j) / \sum_{j=1}^m x_j$, where x_j is the number of transcripts of this gene in cell j when *bimod*=0, x'_j is the number of transcripts of this gene in cell j when *bimod*=1, and m is the number of cells. From Figure S10d we see that there is no obvious increase or decrease of total number of transcripts for the genes when changing *bimod*. So we conjecture the drop in the performance of clustering and DE is rather caused by change in the distribution of gene-expression levels of genes instead of overall gene-expression levels.

We also added corresponding text in the manuscript in Section “Using SymSim to evaluate computational methods for single cell RNA-seq data” and Methods .

Finally, it is common to compare a new method to already existing ones and the authors here choose to compare SymSim to splatter. SymSim produces more realistic datasets than splatter, however Sonesson et al 2018(<http://dx.doi.org/10.1093/bioinformatics/btx631>) show splatter is not the best scRNA-seq available: powsimR (Vieth et al. 2017) simulates much more realistic scRNA-seq than splatter, hence SymSim should be also compared to powsimR.

We thank the reviewer for pointing us to the powsimR. We have strengthened the comparison of SymSim with existing simulators through: 1. adding powsimR to the comparison; 2. adding extra measures to the comparison. We have added Figures S5c-d, S6a-b, S7a-c to show these results. Please refer to our response to Point 4 of Reviewer 2 for details (Page 10 of this document). SymSim shows the best overall performance in terms of all these additional measures compared to published simulators Splatter and powSimR.

Minor comments:

Brownian motion on a tree is not per se new, the authors should cite previous work. For example, there is a whole chapter on this in the Felsenstein book *Inferring Phylogenies* and I believe that it also has been applied in the analysis of RNA-seq data.

We thank the reviewer for pointing out the reference. We have added the mentioned reference (Felsenstein 2004, Chapter 23).

If also the elements of technical variation are estimated, this should be clarified in the text.

SymSim can detect a set of parameter values (out of a large grid) which give the best-match to a real dataset (Supplementary Material Section 4). However, the goal of this procedure is to generate a realistic simulation and we do not claim that the inferred set of parameter values is necessarily the “correct” one, as there can be multiple parameter configurations which result in datasets with similarly (or almost similarly) good match. It is also important to mention that SymSim allows users to generate simulated datasets with different quality by explicitly controlling the values of these technical parameters.

The comparison of the 4 DE-methods is a bit half-hearted (only 4 methods, of which 2 are only applicable to pairwise comparisons + a fairly ad hoc LFC cutoff). It was briefly mentioned that Sonesson et al. 2018 gave similar conclusions. All in all I don't see the added value of the presented DE analysis.

Rather than a comprehensive comparison of DE methods (as done by Sonesson et al), the goal of this section was to showcase the application of SymSim for method benchmarking as a simulator. In particular, SymSim naturally generates DE genes through our framework of EVFs (cell identify vectors), Diff-EVFs (representing biological conditions which cause differences between subpopulations) and gene effect vectors (which defines how much a gene is affected by the Diff-EVFs) (Figure 6a). This results in more realistic DE genes than published simulators (which generate DE genes by altering the gene-expression levels of a set of chosen genes), and the number of DE genes between populations reflect the structure of the tree used to generate the populations (Figure 6c).

What are the run-times and memory requirements for the simulations and the fit the dataset to the database are missing.

We have added runtime and memory requirements for simulating datasets in Methods of the manuscript (Section “Simulation of technical steps from mRNA capturing to sequencing”). Adding technical variation to true counts takes more time than generating the true counts due to multiple sampling steps to mimic the mRNA capturing and PCR amplifications. Fitting a dataset to the database is very efficient as it is a search through the metrics which are readily calculated for all simulated datasets in the database, and the time is linear in terms of the number of simulated datasets.

P.20 I have never seen ‘self-explanatory’ input parameters.

By ‘self-explanatory’ we refer to that the major parameters as input for SymSim have either biological or technical meanings, as opposed to purely distributional models which directly model the gene-expression values. Biological input parameters include: Sigma (σ , representing within-population variation), phyla (the input tree with branch lengths which define the relationships between populations and cross-population variation); technical input parameters include: Alpha_mean (α , mean capture efficiency), Depth_mean (mean sequencing depth). Please refer to Section “Fitting simulation parameters to real data” in Methods and Figure S1 for explanation of more parameters.

At some point the authors start to make general statements about UMI and non-UMI data. If I understood correctly the authors have only analysed 1 UMI and 1 non-UMI data-set and hence this this generalization is not justified.

Importantly, these conclusions are based on simulation experiments, and not an analysis of the two real datasets.

Specifically, we discussed UMI protocol *versus* nonUMI protocol in Paragraph 5 of Section “The third knob: technical variation”. Here we simulated true transcript counts in cells and use respectively UMI and nonUMI protocols to generate observed counts with our simulator. Figure 4c and Figure S4f show that the distribution of observed counts from the UMI protocol is closer to that of true counts than the non-UMI protocol. This discussion is based on simulations and no experimental data is involved. We included the following sentence to clarify: “In Figure 4c, we show the comparison between the simulated true mRNA content of one cell and the simulated observed counts obtained with or without UMI.”

P.20 References for trajectory & RNA-velocity analysis are missing

We have added showcase analysis of benchmarking trajectory inference methods (last paragraph in Section “Using SymSim to evaluate computational methods for single cell RNA-seq data”) and have included references for trajectory inference there. We have included the RNA-velocity reference in Discussion.

Since the authors advertise SymSim to be good for scRNA-seq method evaluation, they should also discuss Ziegenhain et al (2017) who do exactly this using an early version of powsimR.

We thank the reviewer for this reference and added it to this revision.

References

- Dijk, David van, Roshan Sharma, Juozas Nainys, Kristina Yim, Pooja Kathail, Ambrose J. Carr, Cassandra Burdziak, et al. 2018. “Recovering Gene Interactions from Single-Cell Data Using Data Diffusion.” *Cell* 174 (3): 716–29.e27.
- Larson, Daniel R. 2011. “What Do Expression Dynamics Tell Us about the Mechanism of Transcription?” *Current Opinion in Genetics & Development* 21 (5): 591–99.
- Lopez, Romain, Jeffrey Regier, Michael B. Cole, Michael I. Jordan, and Nir Yosef. 2018. “Deep Generative Modeling for Single-Cell Transcriptomics.” *Nature Methods* 15 (12): 1053–58.
- Peccoud, J., and B. Ycart. 1995. “Markovian Modeling of Gene-Product Synthesis.” *Theoretical Population Biology* 48 (2): 222–34.
- Vieth, Beate, Christoph Ziegenhain, Swati Parekh, Wolfgang Enard, and Ines Hellmann. 2017. “powsimR: Power Analysis for Bulk and Single Cell RNA-Seq Experiments.” *Bioinformatics* 33 (21): 3486–88.
- Zappia, Luke, Belinda Phipson, and Alicia Oshlack. 2017. “Splatter: Simulation of Single-Cell RNA Sequencing Data.” *Genome Biology* 18 (1): 174.

Reviewers' comments:

Reviewer #1 (Remarks to the Author):

Summary: The revised manuscript addressed most of my concerns except some minor points.

Minor points:

1. Abstract: Benchmarking trajectory inference methods should be mentioned in the abstract.
2. Methods for trajectory inference: The details of running trajectory inference methods (package version, input parameters, etc) should be clearly described in the Method section.

Reviewer #2 (Remarks to the Author):

- Summary -

This paper describes a simulation strategy for single-cell RNA-seq data using a mechanistic description of the sources of variation and uncertainty in measurement.

In this revised manuscript the authors have added a couple of datasets and alternative methods in comparisons. This places the paper more in the context of typical scRNA-seq experiments and analysis.

The concerns raised in the initial version have mostly been addressed.

It would still be better if the authors did not make use of imputed expression levels. Even though it is common in the field it is not a valid practice, and some commentaries from statisticians have started appearing to indicate of the dangers of this. The authors do explain however that these imputed values are used in a very limited fashion.

The main remaining final concern is regarding the plots comparing real data to simulated data. In the rebuttal the authors explain that $\log(\text{mean} + 1)$ is used. (The legend is changed in the main figures, but it seems the supplementary figures still refer to this as $\log(\text{mean})$). The +1 in these plots are not appropriate, and will (at least in typical 10x data) hide the mean in over 50% of genes with non-zero mean! To see why, make a plot of $\log(\text{mean})$ vs $\log(\text{mean} + 1)$. Genes with mean = 0 can be handled as special cases, e.g. by replacing these values by $\log(1 / (2 * n_cells))$ or similar. This will most likely give a different view on the performance of the different simulators. At the moment only highly expressed genes are represented.

Reviewer #3 (Remarks to the Author):

The authors added many new analyses. Most importantly, they add 10x data, also compare SymSim to powsimR and test the parameter estimation using simulations.

All in all the new analyses make the manuscript much more comprehensive, however they are not discussed critically, but all aspects that do not directly supporting the original storyline appear to be ignored:

In order to match the 10x data the authors added two new parameters to simulate outlier genes that are very highly expressed. These additional parameters appear to be purely empirical and it is unclear

where and how they fit into the estimation procedure described in Figure S1. I think it warrants at least a couple sentences of discussion/speculations why these outliers are not captured by the elaborate generative model that is the heart of SymSim.

In my eyes the estimated distributions of the parameters for the kinetic model in Figure S3a are not a good match for the true simulated parameter distributions. Furthermore the properties for estimated and imputed counts differ quite a lot. For example $\log_{10}(k_{on})$ is bimodal for the imputed counts and unimodal for the true counts. In the response to the reviewers the authors stress that they cannot estimate per cell parameters correctly, but that the distributions are correct. I disagree with this statement and since I do not know which ranges for those parameters are feasible, the statement that the ranges are not completely off is rather weak.

This said, SymSim still simulates datasets that closely mimic the properties of real scRNA-seq data. The authors added more independent statistics to compare SymSim simulations to simulations with splatter and powsimR. In contrast to what a previous independent study by Sonesson et al found, here powsimR appears not even to be able to reproduce the mean counts (Figure S7). I believe this discrepancy is due to powsimR only returning count-matrices after filtering of cells ($>4MAD$ reads/genes detected) and genes (e.g. <0.1 counts on average). This makes sense if we think that these extreme outliers are rather stochastic events due to technical difficulties such as the occasional broken or doublet cell and thus would make parameter estimation more difficult.

In any case, the fact that powsimR does not attempt to reproduce unfiltered count-matrices should be acknowledged somewhere.

Response to Reviewers' Comments

Note:

Responses to comments in the letter below are in blue font

Corresponding edits appear in red font in the revised manuscript

Reviewer #1

Summary: The revised manuscript addressed most of my concerns except some minor points.

Minor points:

1. Abstract: Benchmarking trajectory inference methods should be mentioned in the abstract.

We thank the reviewer for pointing this out. We now mention trajectory inference in the abstract.

2. Methods for trajectory inference: The details of running trajectory inference methods (package version, input parameters, etc) should be clearly described in the Method section.

We have added a section "Applying trajectory inference methods" to the Methods. This new section provides the details of running the trajectory methods.

Reviewer #2

- Summary -

This paper describes a simulation strategy for single-cell RNA-seq data using a mechanistic description of the sources of variation and uncertainty in measurement.

In this revised manuscript the authors have added a couple of datasets and alternative methods in comparisons. This places the paper more in the context of typical scRNA-seq experiments and analysis.

The concerns raised in the initial version have mostly been addressed.

It would still be better if the authors did not make use of imputed expression levels. Even though it is common in the field it is not a valid practice, and some commentaries from statisticians have

started appearing to indicate of the dangers of this. The authors do explain however that these imputed values are used in a very limited fashion.

The main remaining final concern is regarding the plots comparing real data to simulated data. In the rebuttal the authors explain that $\log(\text{mean} + 1)$ is used. (The legend is changed in the main figures, but it seems the supplementary figures still refer to this as $\log(\text{mean})$). The +1 in these plots are not appropriate, and will (at least in typical 10x data) hide the mean in over 50% of genes with non-zero mean! To see why, make a plot of $\log(\text{mean})$ vs $\log(\text{mean} + 1)$. Genes with mean = 0 can be handled as special cases, e.g. by replacing these values by $\log(1 / (2 * n_cells))$ or similar. This will most likely give a different view on the performance of the different simulators. At the moment only highly expressed genes are represented.

Following this comment, we have done the following:

1. Updated figure legends in Supplementary Figures to “ $\log_{10}(\text{mean}+1)$ ” where applicable (Figures S5b-c).
2. Generated Q-Q plots of $\log_{10}(\text{mean})$ to compare the distributions of simulated data by the three simulators (SymSim, Splatter and powsimR) and real data (Figure S5e). We do see differences between distributions of $\log_{10}(\text{mean}+1)$ and $\log_{10}(\text{mean})$, but SymSim is still able to yield reasonably similar distributions of $\log_{10}(\text{mean})$ to that of real data and it does so more accurately than other simulators.
3. Updated text in the manuscript correspondingly (Section “Fitting parameters to real data”).

Reviewer #3

The authors added many new analyses. Most importantly, they add 10x data, also compare SymSim to powsimR and test the parameter estimation using simulations.

All in all the new analyses make the manuscript much more comprehensive, however they are not discussed critically, but all aspects that do not directly supporting the original storyline appear to be ignored:

In order to match the 10x data the authors added two new parameters to simulate outlier genes that are very highly expressed. These additional parameters appear to be purely empirical and it is unclear where and how they fit into the estimation procedure described in Figure S1. I think it warrants at least a couple sentences of discussion/speculations why these outliers are not captured by the elaborate generative model that is the heart of SymSim.

The need to account for high-expression outlier genes through additional parameters has been reported in previous work on simulating single cell RNA seq (Zappia, Phipson, and Oshlack 2017). SymSim was able to account for these genes without specialized parameters in the nonUMI Th17 and the UMI cortex datasets. In 10x data this phenomena is more pronounced, which may be the outcome of selection bias, exacerbated by the low capture rate. We introduced two parameters, *prop_hge* which represent the proportion of these outlier genes, and *mean_hge* which controls how much their gene-expression is augmented. These parameters allow SymSim to simulate datasets for various protocols.

During the process of fitting the parameters to real data, *prop_hge* and *mean_hge* are set in the same way as other major parameters (listed in a table in Section “Fitting simulation parameters to real data” in Methods) using grid search.

We have extended the description in Figure S1 and added discussion into the main text in Section “The first knob: allele intrinsic variation”.

In my eyes the estimated distributions of the parameters for the kinetic model in Figure S3a are not a good match for the true simulated parameter distributions. Furthermore the properties for estimated and imputed counts differ quite a lot. For example $\log_{10}(k_{on})$ is bimodal for the imputed counts and unimodal for the true counts. In the response to the reviewers the authors stress that they cannot estimate per cell parameters correctly, but that the distributions are correct. I disagree with this statement and since I do not know which ranges for those parameters are feasible, the statement that the ranges are not completely off is rather weak. This said, SymSim still simulates datasets that closely mimic the properties of real scRNA-seq data.

To provide information on feasible ranges of the kinetic parameters, we included parameters measured in wet-lab experiments which we obtained from the literature (discussed in Section

“The first knob: allele intrinsic variation”, 5th paragraph, parameters from literature included Supplementary Material Section 1). Following this comment, we have done the following:

1. We improved the clarity of Supplementary Material Section 1 and emphasized the consistency between these data and our simulations. Specifically, we converted the reported values into the same unit (log scale of k_{on} , k_{off} and s over d) as shown in Figure 2b and Figure S3a. We also added a last row into the table with summary of ranges for each parameter. All the ranges are in line with our distributions shown in Figure 2b and Figure S3a.
2. We have included an additional comparison of burst frequency (k_{on}) and burst size (s/k_{off}) using data from a recent study (Larsson et al. 2019). In that paper, Larsson and colleagues inferred burst frequencies and sizes transcriptome-wide (for mice and humans) using allele-sensitive single-cell RNA sequencing. Reassuringly, we show that the ranges reported in that work for mouse fibroblasts are similar to the ones used in our simulation (see Figure S3a-b), while keeping in mind that different cell types can have different burst kinetics (Larsson et al. 2019).
3. Modified main text in Section “The first knob: allele intrinsic variation” to reflect the changes above.

The authors added more independent statistics to compare SymSim simulations to simulations with splatter and powsimR. In contrast to what a previous independent study by Sonesson et al found, here powsimR appears not even to be able to reproduce the mean counts (Figure S7). I believe this discrepancy is due to powsimR only returning count-matrices after filtering of cells (>4MAD reads/genes detected) and genes (e.g. <0.1 counts on average). This makes sense if we think that these extreme outliers are rather stochastic events due to technical difficulties such as the occasional broken or doublet cell and thus would make parameter estimation more difficult.

In any case, the fact that powsimR does not attempt to reproduce unfiltered count-matrices should be acknowledged somewhere.

We have added explanation of this property of powsimR in Section “Fitting parameters to real data” in the main text.

References

- Larsson, Anton J. M., Per Johnsson, Michael Hagemann-Jensen, Leonard Hartmanis, Omid R. Faridani, Björn Reinius, Åsa Segerstolpe, Chloe M. Rivera, Bing Ren, and Rickard Sandberg. 2019. "Genomic Encoding of Transcriptional Burst Kinetics." *Nature* 565 (7738): 251–54.
- Zappia, Luke, Belinda Phipson, and Alicia Oshlack. 2017. "Splatter: Simulation of Single-Cell RNA Sequencing Data." *Genome Biology* 18 (1): 174.

REVIEWERS' COMMENTS:

Reviewer #2 (Remarks to the Author):

In this final revision, all issues have been addressed.

Reviewer #3 (Remarks to the Author):

One of the main properties that supposedly sets Symsim apart from other simulators is that it is based on meaningful parameters.

This would be great, however doubt that the input parameters for the simulations are any more interpretable than the empirical parameters of other simulation programs such as powsimR and splatter.

Furthermore I don't believe that it is ok, to simply copy paste plots from another publication into the supplement, even though it is cited correctly.

Also I did not find the k_{on} / k_{off} ranges in any table

REVIEWERS' COMMENTS:

Reviewer #2 (Remarks to the Author):

In this final revision, all issues have been addressed.

Reviewer #3 (Remarks to the Author):

One of the main properties that supposedly sets Symsim apart from other simulators is that it is based on meaningful parameters.

This would be great, however doubt that the input parameters for the simulations are any more interpretable than the empirical parameters of other simulation programs such as powsimR and splatter.

We thank the reviewer for making this comment. First, we would like to stress that several of the parameters, especially the ones at the third knob are interpretable and provide a way to investigate modulation of important experimental factors. We acknowledge this is not the case for all the parameters though, and as a response to this comment we now discuss the possible future directions to mitigate this in the discussion section. For example, generative models like scVI (Lopez et al., 2018) can be used to replace our first and second knobs, which reduces the extent of parametrization.

Furthermore I don't believe that it is ok, to simply copy paste plots from another publication into the supplement, even though it is cited correctly.

We have removed that plot, as suggested in Editorial Requests, and referred to the original paper instead.

Also I did not find the k_{on} / k_{off} ranges in any table

This data is provided in Supplementary Table 1.